# Subtle selectivity in a pheromone sensor triumvirate desynchronizes competence and predation in a human gut commensal

Johann Mignolet[1,2‡*], Guillaume Cerckel[1†], Julien Damoczi[1†], Laura Ledesma-Garcia[1], Andrea Sass[3], Tom Coenye[3], Sylvie Nessler[4], Pascal Hols[1]

[1]Biochemistry and Genetics of Microorganisms (BGM), Louvain Institute of Biomolecular Science and Technology, Université catholique de Louvain, Louvain-la-Neuve, Belgium; [2]Syngulon, Seraing, Belgium; [3]Laboratory of Pharmaceutical Microbiology, Ghent University, Ghent, Belgium; [4]Institute for Integrative Biology of the Cell (I2BC), CEA, CNRS, Univ. Paris-Sud, Université Paris-Saclay, 91198, Gif-sur-Yvette cedex, France

*For correspondence:
johann.mignolet@uclouvain.be;
jmignolet@syngulon.com

[†]These authors contributed equally to this work

Present address: [‡]Syngulon, Seraing, Belgium

**Abstract** Constantly surrounded by kin or alien organisms in nature, eukaryotes and prokaryotes developed various communication systems to coordinate adaptive multi-entity behavior. In complex and overcrowded environments, they require to discriminate relevant signals in a myriad of pheromones to execute appropriate responses. In the human gut commensal *Streptococcus salivarius*, the cytoplasmic Rgg/RNPP regulator ComR couples competence to bacteriocin-mediated predation. Here, we describe a paralogous sensor duo, ScuR and SarF, which circumvents ComR in order to disconnect these two physiological processes. We highlighted the recurring role of Rgg/RNPP in the production of antimicrobials and designed a robust genetic screen to unveil potent/optimized peptide pheromones. Further mutational and biochemical analyses dissected the modifiable selectivity toward their pheromone and operating sequences at the subtle molecular level. Additionally, our results highlight how we might mobilize antimicrobial molecules while silencing competence in endogenous populations of human microflora and temper gut disorders provoked by bacterial pathogens.
DOI: https://doi.org/10.7554/eLife.47139.001

## Introduction

In the living world, all organisms are parts of multi-species ecosystems. Some niches such as the human digestive tract are densely populated with more than 1000 species interacting with each other through competition or cooperation for nutrients and colonization areas (*Huang et al., 2011*; *Kommineni et al., 2015*; *The Human Microbiome Project Consortium, 2012*). Therefore, metazoan and unicellular organisms developed social skills and set up kin, interspecies and even interkingdom trans-communication via pheromones (*Hughes and Sperandio, 2008*; *Kholodenko, 2006*; *Waters and Bassler, 2005*). This kind of behavior favors the coordination of cellular processes to provide a concerted response. In bacteria, this results in the production of defense/assault molecules targeting microorganisms or host immune cells, sporulation, mass locomotion, biofilm formation or acquisition of new genetic material (*Waters and Bassler, 2005*).

For intraspecies communication, so-called quorum sensing (QS), Gram-negative bacteria usually signal through secondary metabolites, for example homoserine lactone or auto-inducers II (*Papenfort and Bassler, 2016*). Otherwise, Gram-positive bacteria secrete ribosomally-produced peptides that are detected by two distinct mechanisms. In case of two-component systems (TCSs), the small extracellular peptide canonically contacts a membrane histidine kinase that conveys the

phosphate-based information to a transcription factor (*Kleerebezem et al., 1997*). Alternatively, a diffusible peptide can be internalized through the general oligopeptide transporter Opp (or Ami) and can bind a cytoplasmic receptor of the RRNPP family to modulate transcription of specific genes (*Cook and Federle, 2014*; *Neiditch et al., 2017*). The RRNPP regulators (stand for the archetype proteins Rgg, Rap, NprR, PlcR and PrgX) are ubiquitous in Firmicutes. They harbor a typical tetratricopeptide repeat (TPR) domain, which docks the pheromone and, apart from the Rap phosphatases, an N-terminal helix-turn-helix (HTH) domain that recognizes specific DNA stretches to turn on target promoters (*Grenha et al., 2013*; *Neiditch et al., 2017*; *Zouhir et al., 2013*).

In the *Streptococcus* genus, foreign gene acquisition through accumulation of the master regulator of competence for natural transformation ComX (also known as SigX) is dictated by social abilities (*Fontaine et al., 2015*). Whereas the signaling cascade in the anginosus and mitis groups (that include *Streptococcus pneumoniae*) is based on external sensing via the ComC-responsive ComDE TCS (*Martin et al., 2013*; *Pestova et al., 1996*), all other streptococci (mutans, bovis, pyogenes and salivarius groups) rely on an intracellular Rgg/RNPP-small hydrophobic peptide tandem known as the ComRS system (*Figure 1A*) (*Fontaine et al., 2013*; *Mashburn-Warren et al., 2010*; *Mignolet et al., 2018*). The ComS pheromone is basally produced and concomitantly exported/matured to accumulate as a XIP (comX/sigX-inducing peptide) form in the extracellular medium. Then, it penetrates the intracellular compartment by non-specific translocation (Opp import system) and docks with the peptide-binding pocket of the ComR TPR domain. Subtle reorganizations in the TPR domain conformation fracture the solenoid-fold of the α-helix nine in a newly described mode of activation for RRNPP members (*Shanker et al., 2016*; *Talagas et al., 2016*). In the peptide-free ComR, this helix sequesters specific arginine residues of the HTH domain, preventing interaction with the major groove of DNA backbone. Therefore, the α-helix nine break liberates the HTH domain from the TPR domain grip and facilitates ComR dimerization. The ComR•XIP binary complex binds the ComR-box in the *comS* promoter to robustly initiate a positive feedback loop. In parallel, it also binds the ComR-box of the *comX* promoter (*Mignolet et al., 2018*), which ultimately drives biogenesis of the transformation machinery (transformasome). Furthermore in *S. salivarius*, the ComR•XIP complex directly turns on bacteriocin gene promoters to link competence and predation (*Mignolet et al., 2018*). This contrasts with all other streptococci for which the BlpRH TCS is the QS system regulating bacteriocin production in a self-sufficient manner, even though its activity could be modulated by ComR or ComDE (*Shanker and Federle, 2017*). Such a predation-competence coupling mechanism presumably guarantees the competent cells that the killing effect of toxins liberates genetic material from dead sensitive cells (*Veening and Blokesch, 2017*). However, it could be regarded as a risky strategy that prevents the release of the bacterial arsenal during conditions inappropriate for entry into the competence state.

Here, we describe a new communication system that instates predation independently of competence in *S. salivarius*, thus restoring one degree of freedom on bacteriocin production. With phenotypical, biochemical and deep-sequencing approaches, we unveiled that ScuR, a ComR-like RRNPP, regulates the production of salivaricins, but not ComX, due to a strict and sophisticated selectivity of ComR-box recognition. In parallel, we developed a genetic screen to identify optimized/synthetic pheromones for ScuR and potentially cytoplasmic peptide sensors in general. The conservation of ScuR in the *S. salivarius* clade suggests that it fulfills the function adopted by BlpRH in other streptococci. It also underlines the contrast between the predominant role of RRNPPs in *S. salivarius* vs TCSs in *S. pneumoniae* on predation-competence (un)coupling. Finally, the ScuR pathway is a tantalizing target to hijack in order to mobilize bacteriocins in the human microflora and minimize horizontal gene exchanges (*Hols et al., 2019*).

## Results

### Regulon interweaving in ComR paralogs

The direct regulation of bacteriocins by ComR in the *S. salivarius* species is uncommon and suggests a positive selection for species-specific strategies that participate in niche adaptation (*Mignolet et al., 2018*). Interestingly, the *S. salivarius* HSISS4 genome encodes five RRNPP transcriptional factors, including ComR. Two regulators renamed ScuR (HSISS4_01166; stands for salivaricins-competence uncoupling regulator) and SarF (HSISS4_01169; ScuR-associated Rgg factor) for reasons

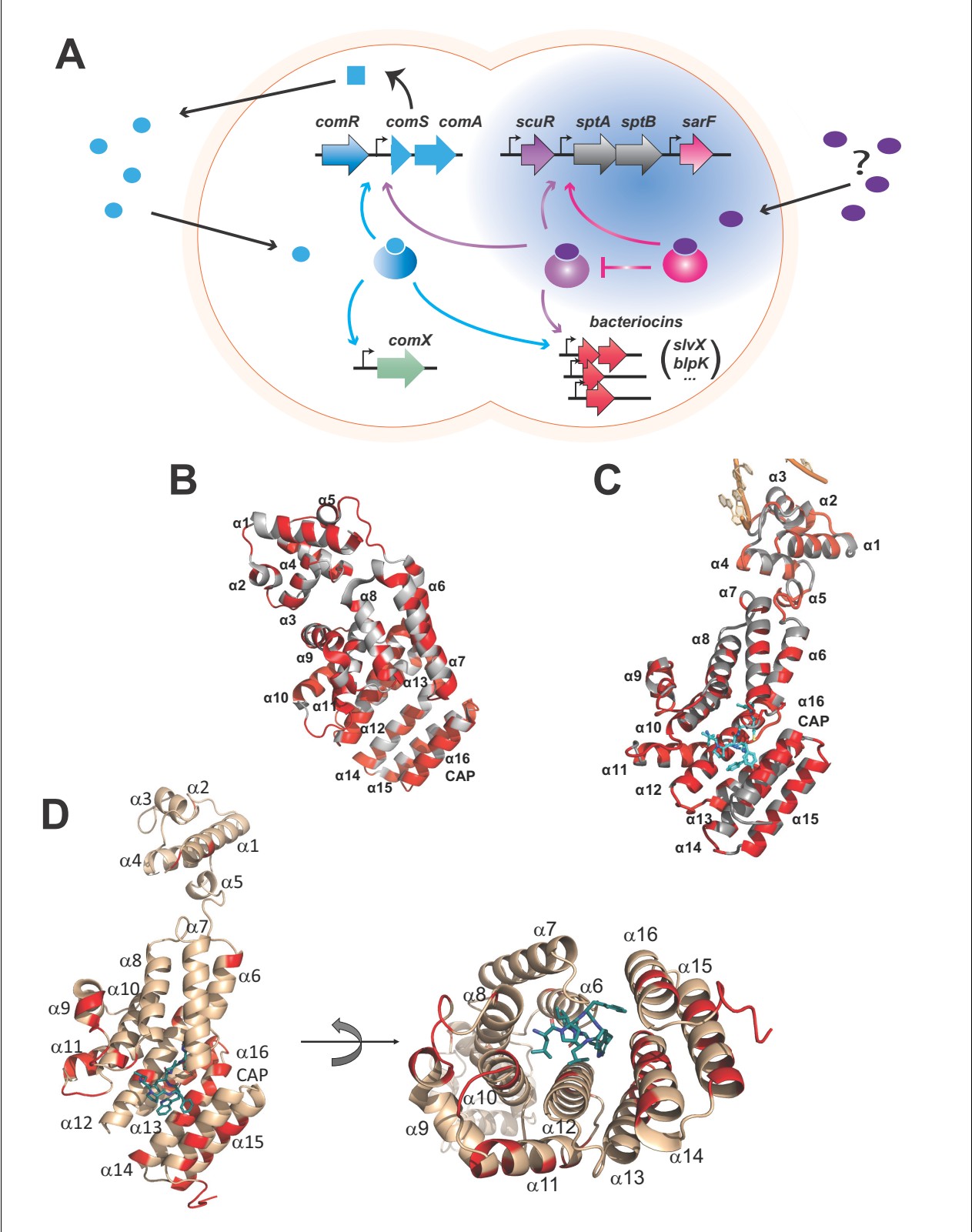

**Figure 1.** Rgg-based decision for competence-predation activation in *S. salivarius*. (**A**) Scheme of genomic organization and transcriptional dependencies (color-coded arrows) between competence activation (*comX*) and bacteriocins production (*blpK, slvX, . . .*) in *S. salivarius*. Promoters are depicted with broken arrows. Regulators (large gradient-colored ellipses) and the ComS pheromone (uniform blue shapes) are colored according to their encoding genes. The ComS precursor is produced (curled plain arrow) as an intracellular precursor (blue square) before secretion, maturation and

*Figure 1 continued on next page*

*Figure 1 continued*

import as an active pheromone (blue ellipses). The newly described two-Rgg system (*Figure 1—figure supplement 1*) is highlighted in blue and the T arrow pinpoints the inhibitory role of SarF on ScuR. Their cognate pheromone of unknown origin is purple-colored. (**B**) Crystal structure of the monomeric apo form of *S. thermophilus* ComR (PDB ID 5JUF). The protein is shown as cartoon colored in gray. The HTH and TPR domains are labeled, as well as the linker region. Residues which are not conserved in ScuR are highlighted in red. The numbering of the α-helices is indicated to gain clarity. (**C**) Crystal structure of *S. thermophilus* ComR in complex with sComS (LPYFAGCL) (PDB ID 5JUB). Only one ComR subunit of the dimeric ComR•sComS•DNA complex is shown in gray with ScuR substitution sites in red as in (**B**). The bound peptide is highlighted in blue sticks. Part of the bound DNA is shown in orange and labeled, as the HTH and TPR domains. (**D**) Orthogonal views of the ScuR-sBI7 model. The protein is shown as cartoon colored in beige except for residues substituted in SarF, which are highlighted in red as in *Figure 8C*. The bound peptide is shown in blue sticks with the conserved tryptophan (W). The numbering of α-helices is indicated to gain clarity. In the right panel, the HTH domain is hidden in the back of the figure. The model was obtained by homology modeling using the i-TASSER server and the ComR•sComS•DNA complex (PDB ID 5JUB) as template.

DOI: https://doi.org/10.7554/eLife.47139.002
The following figure supplement is available for figure 1:

**Figure supplement 1.** Sequence alignments of ComR paralogs.
DOI: https://doi.org/10.7554/eLife.47139.003

detailed below share a high level of identity with *S. salivarius* ComR (42%) (*Figure 1—figure supplement 1A*), contrasting with the two other RRNPP proteins that do not share any primary sequence similarity. Whereas the amino-acids involved in dimerization ($D_{200}$-$K_{87}$ and $K_{246}$-$E_{282}$ pairs) and pheromone selectivity ($P_{89}T_{90}Y_{91}R_{92}$ motif and $S_{248}$) are partially conserved in ScuR and SarF, the residues responsible for the HTH sequestration in ComR (*Talagas et al., 2016*) are well-conserved in both regulators, suggesting that these homologs could share a similar mode of activation (*Figure 1B and C* and *Figure 1—figure supplement 1B*). Moreover, the paralogous ScuR and SarF proteins are highly similar (82% identity). Strikingly, residue divergences are nearly all concentrated in only three amino acid stretches, one of which overlaps the α-helix 14 that forms part of the peptide recognition pocket (*Figure 1D* and *Figure 1—figure supplement 1B*). This indicates that the two proteins could likely accommodate specific and different peptides. On the chromosome, the *scuR* and *sarF* genes are located in the same locus, separated by two genes that code for two predicted subunits of an ABC transporter, SptA and SptB (for <u>S</u>cuR-<u>p</u>romoted <u>t</u>ransporter <u>A</u> and <u>B</u>, respectively) (*Figure 1A*). In contrast to characterized *rgg/comR* loci, no short coding sequence was distinguishable upstream or downstream of *scuR* and *sarF* genes, indicating a different genomic coding topology of the communication system.

Due to the extensive conservation between ComR, ScuR and SarF, especially in the DNA binding domain (*Figure 1B and C* and *Figure 1—figure supplement 1A and B*), we questioned whether the two uncharacterized paralogs are capable to control competence and predation as well. Hence, we extracted mRNA of the wild-type (WT) strain and of engineered in-frame deletion mutants (Δ*scuR* and Δ*sarF*) and quantified gene expression via deep sequencing (RNAseq). With no hint about the genuine activating pheromones, we also included overexpression mutants (*scuR$^{++}$* and *sarF$^{++}$*) in our high-throughput transcriptome analyses. Indeed, a strong overproduction of ComR was reported to be sufficient for activation of its target promoters, even in absence of ComS (from endogenous production or synthetic peptide addition) (*Mignolet et al., 2018*). Both deletion mutants did not have a dramatically altered transcriptome compared to the WT strain (*Supplementary file 2*), meaning that ScuR and SarF have only a minor function during standard growth conditions. However, the loss of SarF slightly increased *scuR* expression, while the *sptA* and *sptB* mRNA level almost increased 5-fold, suggesting that SarF could be a repressor/antagonist of the ScuR-SptAB system. In contrast, the strong overexpression of *scuR* (28-fold) elicited a very strong activation of the *sptA-sptB* operon (about 2000-fold). Furthermore, a second cluster of genes, all located inside salivaricin loci, was robustly upregulated, although with a lower magnitude (ranging from 35- to 140-fold) (*Supplementary file 3*). Surprisingly, *comX* mRNA levels remained approximately stable in all mutants. Altogether, these results imply that the ComR, ScuR and SarF paralogs might control overlapping but dedicated regulatory networks.

## ScuR is an alternative self-sufficient sensor that controls salivaricin production but keeps competence off

In order to validate our transcription profile analyses, we performed promoter-probe assays, as previously described for ComR (*Mignolet et al., 2018*). We first expanded our collection of luciferase reporter strains (composed of *comS*, *comX* and bacteriocin gene promoters) to include and monitor the *sptA* promoter (P$_{sptA}$), and next transformed all of them with a *scuR* overexpression cassette. Finally, we measured promoter activity in presence or absence of sComS (synthetic octapeptide that corresponds to the shortest active form of XIP: LPYFAGCL) during cell growth (*Figure 2A*). In agreement with our RNA-seq data, *sptA* and bacteriocin promoters were all markedly up-regulated in the *scuR*$^{++}$ strain, irrespective of the addition of sComS and with no significant synergy. In addition, despite the high conservation between ComR and ScuR HTH domains, we observed no activation of P$_{comX}$ due to ScuR accumulation, ruling out ScuR as a trigger of competence. Nonetheless, the P$_{comS}$ showed a 35-fold change in activity, suggesting that ScuR might modulate ComR cell-signaling. To strengthen our understanding of this bipartite system, we assessed the activity of P$_{sptA}$, P$_{comS}$ and P$_{slvX}$ in a *sarF*$^{++}$ strain, and noticed that *scuR* and *sarF* overexpression governs P$_{sptA}$ activation amplitude in a similar range, while P$_{comS}$ and P$_{slvX}$ are irresponsive to SarF (*Figure 2B*).

Considering that the effects of transcriptional regulators could be indirect, we inactivated individually the three Rggs by gene deletion in our reporter strains and determined the residual activity of the others. We discovered that ScuR still controls both P$_{sptA}$ and P$_{comS}$ in *comR* (*Figure 2C*) or *sarF* deleted strains (*Figure 2D*), while SarF is sufficient to activate P$_{sptA}$ even in absence of *scuR* (Δ*scuR-sarF/sarF* $^{++}$ mutant) (*Figure 2D*). Finally, the sComS-mediated regulation of ComR is not diminished in absence of both ScuR and SarF (*Figure 2—figure supplement 1*), suggesting that each regulator can stand alone to fulfill its function and work in parallel.

Taken together, our results suggest that the three transcriptional factors have partial redundant functions, even if they harbor regulon specificities, presumably to ensure a broader diversity of cellular response to environment stresses. ScuR and SarF, but not ComR, control the *sptAB* operon, while ScuR alone has a ComR-independent extra regulatory role on bacteriocin production. Even though ScuR promotes ComS production, this regulator does not act on the *comX* promoter and is likely to disconnect the competence-predation coupling compelled by ComR (*Mignolet et al., 2018*).

## Randomization-based screen for pheromone identification

Typically, the major challenge in characterizing the transduction mechanism of cell-cell communication sensor is the identification of the ligand(s) or the perceived signal(s). As inspection of the genome did not reveal any small peptide encoded in the vicinity of the *scuR-sarF* locus, we decided to conduct a screen to identify synthetic peptides able to activate the ScuR/SarF system (*Figure 3A*). We first constructed a strain harboring a translational fusion of the ScuR/SarF-specific P$_{sptA}$ to a gene conferring resistance to chloramphenicol (*cat*). We then amplified a DNA fragment that allows recombination at a permissive locus (tRNA$^{Ser}$) and encompasses, under xylose control, a 12 codons-long nucleotide sequence, the last 7 of which are randomized (see Materials and methods). We finally transformed this PCR product into the above-mentioned reporter mutant and selected clones on plates supplemented with chloramphenicol and xylose (0.1 or 1%). Note that, in order to increase the transformation rate or decrease the cytotoxicity due to concomitant bacteriocin production, we worked in *comR* overexpression (P$_{xyl1}$-*comR*) or salivaricin deprived (Δ*slv5*) backgrounds, respectively. In total, after nine independent screening runs, we collected around a hundred clones that we streaked again on selective medium with and without xylose. Clones that showed a visual improvement of growth in the presence of chloramphenicol and xylose (*Supplementary file 4*) were sent for sequencing. As a negative control, we included a clone (BM1) with xylose-independent growth in our further analyses. Out of the 30 positive clones, 22 harbored a non-redundant peptide/nucleotide sequence. In order to discard clones with secondary mutations for which the survival phenotype was not related to the peptide nature, we amplified for each clone the complete locus that encodes the small dodecapeptide and backcrossed it into WT or Δ*slv5* backgrounds. We then confirmed on solid media that chloramphenicol resistance qualitatively increased upon xylose addition (*Figure 3B*). We used the same PCR products to transform a strain harboring the P$_{sptA}$-*luxAB* reporter fusion and quantify the influence of peptide production. Again, we noticed that xylose addition potentiated the

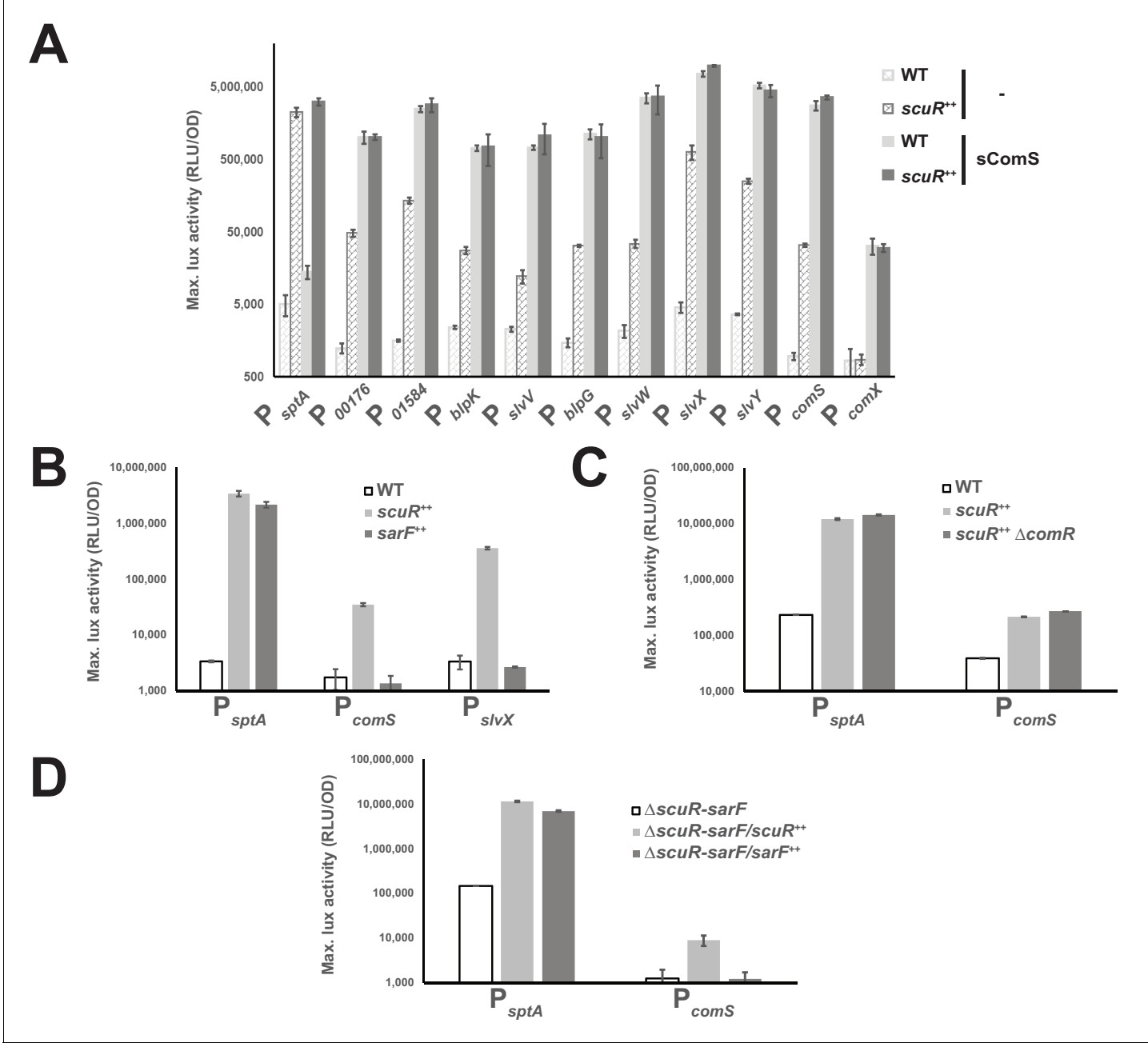

**Figure 2.** Competence-predation desynchronization in *S. salivarius*. (A, B, C and D) Maximum luciferase activity/OD$_{600}$ ratio (RLU/OD; logarithmic scale) of various promoters involved in competence or bacteriocin production fused to a *luxAB* reporter system in WT or overexpressing backgrounds (*Figure 2—figure supplement 1*). (A) Promoter activation of genes upon sComS addition (full bars) vs mock condition (striped bars) in WT (light gray bars) or *scuR* overexpression mutant (*scuR*$^{++}$; dark gray bars). (B) Activity of *sptA*, *comS* and *slvX* promoters in WT strain, and scuR (*scuR*$^{++}$) or sarF (*sarF*$^{++}$) overexpression mutants. (C) Activity of *sptA* and *comS* promoters in WT and *scuR*$^{++}$ mutant deleted or not of *comR* gene. (D) Activity of *sptA* and *comS* promoters in a Δ*scuR-sarF* background when *scuR* or *sarF* are overexpressed. Experimental values represent averages (with standard error of the mean, SEM) of at least three independent biological replicates.

DOI: https://doi.org/10.7554/eLife.47139.004

The following figure supplement is available for figure 2:

**Figure supplement 1.** The ComS-driven activation is ScuR/SarF-independent.

DOI: https://doi.org/10.7554/eLife.47139.005

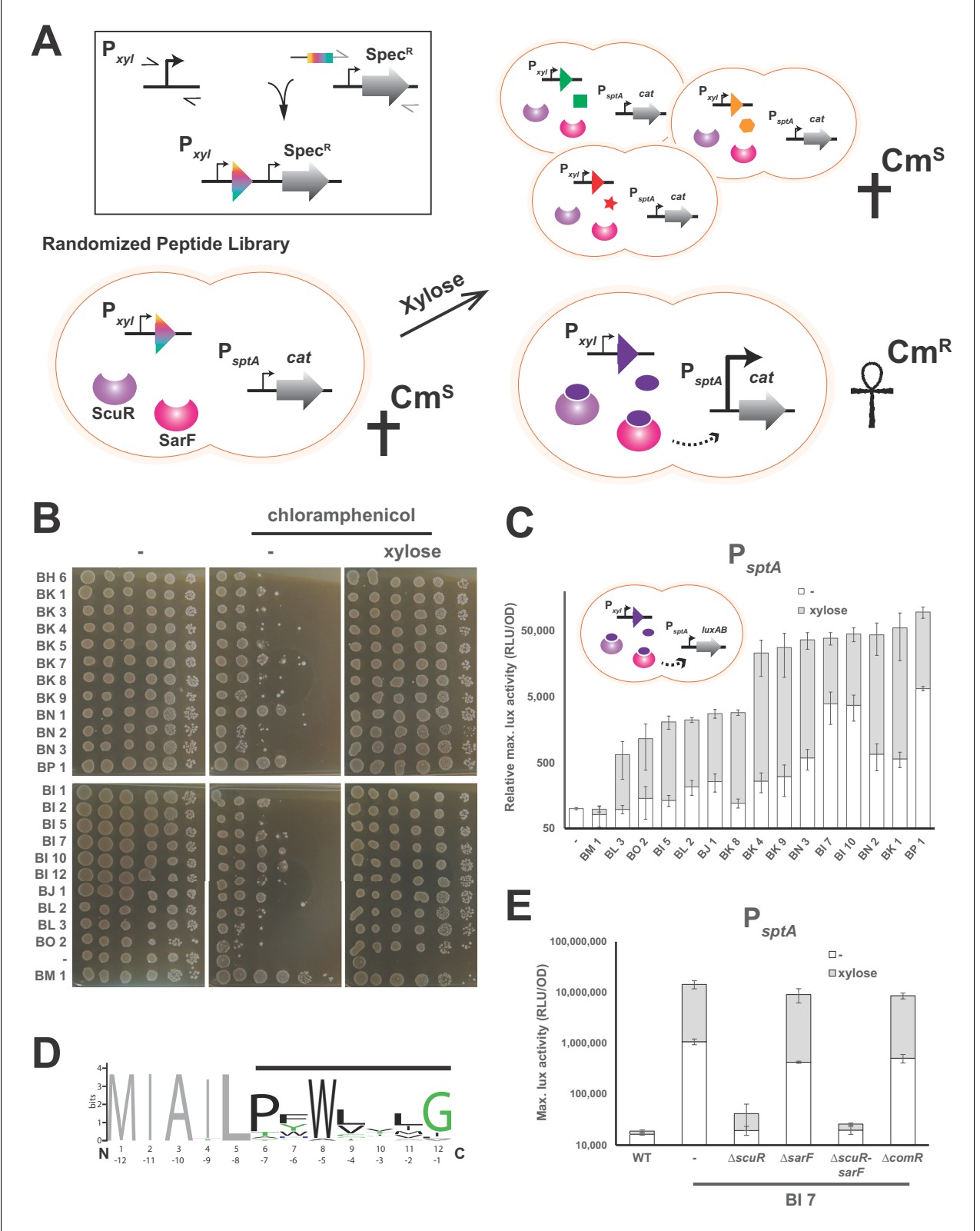

**Figure 3.** Identification of ScuR/SarF activating peptide. (**A**) Cartoon portraying the rational strategy for the peptide randomization-based screen. In the box on the top-left: a degenerated primer (rainbow box) was used to amplify two kinds of xylose-inducible promoters, P$_{xyl1}$ (strong and leaky) and P$_{xyl2}$ (mild and tight). The 5' end of this primer hybridizes on a second amplicon encompassing a spectinomycin resistance cassette (Spec$^R$). Both PCR products were grafted with an overlapping PCR. Next, a library of randomized small genes (rainbow arrow head) under inducible promoter control (P$_{xyl}$)

*Figure 3 continued on next page*

*Figure 3 continued*

is transformed into a reporter strain in which the chloramphenicol resistance gene (*cat*) is translationally fused to *sptA* promoter. In absence of xylose or upon xylose induction of irrelevant peptides (green square, orange hexagon and red star), *sptA* promoter remains OFF and does not initiate *cat* transcription, causing cell sensitivity (Cm$^S$) on chloramphenicol-supplemented media. The xylose-driven intracellular production of a cognate peptide (purple ellipse) promotes chloramphenicol resistance (Cm$^R$), through P$_{sptA}$ activation by ScuR/SarF (dashed arrow). (**B**) Viability test of WT and all non-redundant mutants expressing intracellularly activating peptides in the P$_{sptA}$-*cat* reporter background. The BM1 clone was used as an irrelevant peptide (negative control). Overnight precultured cells were diluted (OD$_{600}$ of 0.05) to inoculate fresh M17G medium and grown 3 hr (OD$_{600}$ of 0.5). Before plating on medium supplemented or not with chloramphenicol (2 mg.ml$^{-1}$) and/or xylose (1% top panel; 0.1% bottom panel), the culture was sampled and serially diluted (10:10) in M17G. (**C**) Activity of *sptA* promoter in WT strains and various mutants intracellularly expressing activating peptides (cartoon) in medium supplemented with xylose (0.1% or 1%; gray bars) vs mock conditions (open bars). The BM1 clone is an irrelevant peptide (negative control). Magnitude is expressed in percentage compared to the WT P$_{sptA}$-*luxAB* reporter strain (Relative maximal luciferase activity). Experimental values represent the averages (with standard error of the mean, SEM) of at least three independent biological replicates. (**D**) Weighted consensus sequence for 22 activating peptides identified in the randomization-based screen (*Figure 3—figure supplement 1*). Randomized residues are highlighted with a horizontal black bar while the non-variable amino acids are gray-coloured. The Bits represent the relative frequency of residues. Information content is plotted as a function of residues position and depicted from the N-terminus (1 to 12) or the C-terminus (−1 to −12). The sequence logo image was generated using the WebLogo application (http://weblogo.berkeley.edu/logo.cgi). (**E**) Promoter activity of the *sptA* gene in response to the BI7 encoded peptide in various *scuR*, *sarF* or *comR* mutant backgrounds. Media were supplemented with 0.1% xylose (open bars) or water (gray bars). Experimental values represent averages (with standard error of the mean, SEM) of at least three independent biological replicates.

DOI: https://doi.org/10.7554/eLife.47139.006

The following figure supplement is available for figure 3:

**Figure supplement 1.** Sequence conservation of ScuR/SarF activating peptides.
DOI: https://doi.org/10.7554/eLife.47139.007

promoter activity with values ranging from 5 to 100 fold, while it had no effect on BM1 (negative control) and WT strains (*Figure 3C*).

We aligned the 22 unique peptide sequences to find common chemical properties (*Figure 3D* and *Figure 3—figure supplement 1*). Strikingly, a tryptophan residue was highly conserved at position −5 from the C-terminus. On the top of this, the adjoined position (−6) was mainly occupied by an aromatic residue and a proline was mainly observed at position −7. Finally, the position −1 was preferentially a glycine. Positions −2,–3 and −4 varied more, although we observed a tendency towards hydrophobic amino acids. Altogether, this is reminiscent of ComS pheromones extruded by streptococci but surprisingly appears to be a hybrid between type I ComS from salivarius group (two aromatic residues at positions −5 and −6) and type II ComS from mutans, bovis and pyogenes groups (a conserved W and a G at C-terminus) (*Fontaine et al., 2015*). Given that it encoded a peptide (MIAILPFWLILG) that neatly mimicked the consensus sequence (MIAILPFWLVLG), we decided to focus on the clone BI7, and we found that ScuR was specifically responsible for the xylose-driven phenotype. Indeed, neither *comR* nor *sarF* deletion had a dramatic effect, while ScuR loss annihilated both xylose induction and basal leaky expression (*Figure 3E*).

## Exogenous synthetic pheromones selectively activate the ScuR-SarF pair

Likewise ComS and ComR, we next checked whether the ScuR-SarF system could be activated with synthetic peptides (*Figure 4A*). We therefore selected a representative panel of peptides from our screen, ordered the synthesis of the last eight amino acids, and tested P$_{sptA}$ activation (*Figure 4B*). Whatever their degree of kinship toward the consensus motif, all peptides were capable of inducing light production when added to the medium. However, a weaker activation was displayed by the peptides that diverge the most from the consensus sequence such as sBK3, which does not harbor a C-terminus glycine, or sBK4, for which the tryptophan and glycine are shifted by one position (exacerbated effect at the non-saturating concentration of 0.01 μM). The huge variability in sequence and the similar amplitude of activation for all other peptides emphasizes that properties of the residues between the conserved tryptophan and glycine and the proline (position −7) are not essential for ScuR or SarF transactivation, while substitutions at the position −6 are tolerated as long as the amino acid nature is aromatic. Moreover, directed mutations of the conserved tryptophan demonstrated the absolute requirement of the indole moiety, considering that neither alanine (sBI7$^{W \to A}$) nor phenylalanine (sBI7$^{W \to F}$) variants sustained luciferase transcription at low peptide concentration (0.001 μM; *Figure 4C*). A similar strategy for the C-terminal glycine showed that substitution by an

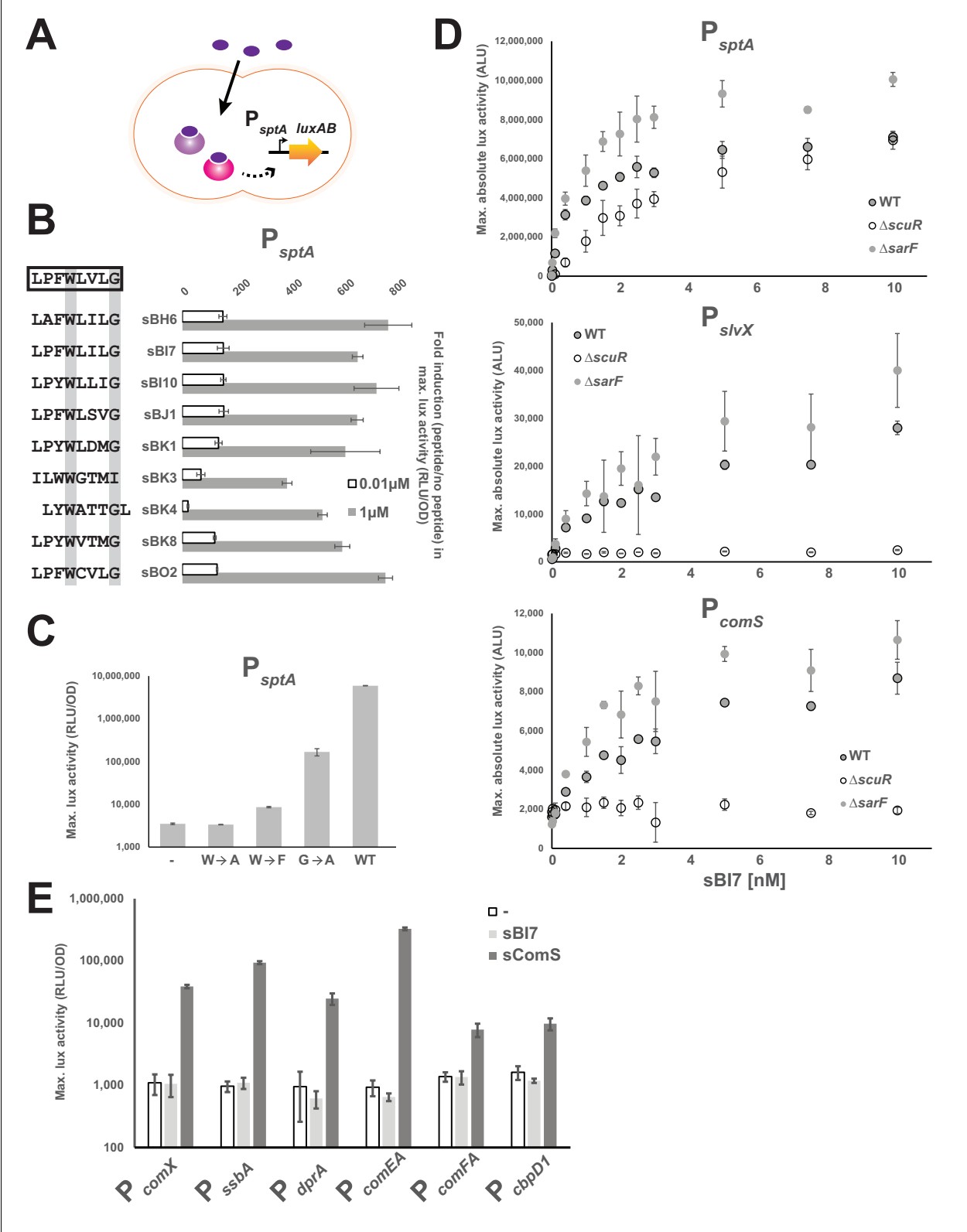

**Figure 4.** The ScuR/SarF system responds to exogenous peptides. (**A**) Cartoon depicting the ScuR/SarF-mediated activation of P$_{sptA}$ upon addition of exogenous synthetic peptides. (**B**) Fold increase in maximal P$_{sptA}$ activity upon addition of representative synthetic peptide (0.01 or 1 μM) vs mock conditions. Peptide sequences are associated to peptide names and compared to the consensus motif (open box). The highly conserved W and G residues are highlighted with gray boxes. (**C**) Maximal activity of P$_{sptA}$ exposed to WT and mutant sBI7 peptides (1 nM) (***Figure 4—figure supplement***

*Figure 4 continued on next page*

*Figure 4 continued*

1). (D) Dose response dot plot of *sptA*, *slvX* and *comS* promoter activity upon sBI7 induction at various concentrations in nM (maximal absolute luciferase activity). Promoters were tested in WT strain and Δ*scuR* or Δ*sarF* mutants (*Figure 4—figure supplement 2*). (E) Maximal activity of P$_{comX}$ and ComX-dependent promoters exposed to the sComS (light gray) or sBI7 (dark gray) peptide (1 μM) in comparison to basal activity (open box) (*Figure 4—figure supplement 3*). (B, C, D, and E) Experimental values represent averages (with standard error of the mean, SEM) of at least three independent replicates. Some standard errors are too small to be visualized in (B), (D), and (E).

DOI: https://doi.org/10.7554/eLife.47139.008

The following figure supplements are available for figure 4:

**Figure supplement 1.** Amino acid requirements for the sBI7-mediated effect.
DOI: https://doi.org/10.7554/eLife.47139.009
**Figure supplement 2.** Loss of ScuR/SarF, but not ComR, annihilates the sBI7-mediated effect.
DOI: https://doi.org/10.7554/eLife.47139.010
**Figure supplement 3.** sBI7 has no effect on the *comX* promoter activation.
DOI: https://doi.org/10.7554/eLife.47139.011

alanine (sBI7$^{G→A}$) decreases the P$_{sptA}$ response although to a lesser extent compared to the tryptophan (*Figure 4C*). It is noteworthy that high concentration of sBI7$^{W→F}$ and sBI7$^{G→A}$ (but not sBI7$^{W→A}$) can bypass the requirement of the tryptophan and glycine and activate P$_{sptA}$ in a similar range than the WT sBI7 peptide (*Figure 4—figure supplement 1*), suggesting that the mutations do not totally abrogate the ScuR/SarF activation but rather modulate the dynamics of interaction.

To refine our understanding of the two Rgg systems, P$_{sptA}$, P$_{comS}$ and P$_{slvX}$ were challenged with increasing amounts of sBI7 at low concentration in WT, Δ*scuR* or Δ*sarF* strains (*Figure 4D*). In line with our overexpression data, P$_{comS}$ and P$_{slvX}$ were totally insensitive to SarF (no activity in Δ*scuR*), while both ScuR and SarF could turn on P$_{sptA}$ independently of each other. Furthermore, in WT backgrounds, we observed that all promoters were responsive to less than 1 nM of peptide, with the highest amplitude for P$_{sptA}$ and the lowest for P$_{comS}$. Activity of all promoters in Δ*sarF* was slightly higher compared to the WT, supporting the notion that SarF might have a mild inhibitory effect on ScuR function. As expected, the induction provoked by sBI7 addition was abolished in a Δ*scuR-SarF* double mutant (*Figure 4—figure supplement 2A*), however sBI7 was surprisingly able to induce the SarF-mediated P$_{sptA}$ response (*Figure 4D*). This discrepancy with regard to the results for the genome-encoded peptide (*Figure 3E*) might be due to inherent differences imposed by the screening method compared to the exogenous supplementation of a synthetic peptide (e.g. variable intracellular concentration, different peptide length, lower activation rate of SarF). However, it underlines that the peptide-binding pocket of both Rgg could accommodate a specific pheromone that does not interfere with the ComR signaling pathway (*Figure 4—figure supplement 2B*). Next, we dissected the relative contributions of ScuR and SarF in *sptA* and salivaricin gene trans-activation (*Figure 4—figure supplement 2C*). In agreement with our *scuR* overexpression data (*Figure 2A*), we first observed that P$_{slvX}$ and P$_{slvY}$ are the most reactive bacteriocin promoters compared to P$_{blpK}$, P$_{slvV}$ and P$_{slvW}$. Next, we recapitulated the two kinds of promoter classes. On the one hand, only the double *scuR-sarF* deletion (but not individual mutants) markedly impinges on P$_{sptA}$ activation. On the other hand, the bacteriocin promoters displayed a weaker and similar activation in both single *scuR* deletion and double *scuR-sarF* deletion. Finally, we confirmed that sBI7 cannot activate *comX* and late competence genes under direct ComX control (*Figure 4E*), and is ineffective in supporting natural transformation (*Table 1*), whereas ScuR and SarF have no effect on the sComS-mediated competence entry (*Figure 4—figure supplement 3*). Altogether, these results reemphasize the predominant role of ScuR in the dedicated bacteriocin production and the complementary role of ScuR and SarF on SptA synthesis.

In order to evaluate selectivity and affinity of synthetic peptides toward their cognate sensors, we determined direct interaction through fluorescent polarization assays, a successful low-volume technique particularly adapted for small ligands (*Moerke, 2009*). To do this, we titrated a fixed concentration of fluorophore-conjugated peptides (sBI7 or sComS) with increasing amounts of purified proteins (ScuR, SarF or ComR). Anisotropy measurements revealed that ScuR and ComR exhibit a similar range of binding for sBI7 and sComS, respectively, with an affinity factor (EC$_{50}$) approximatively seven times lower for the ScuR•sBI7 couple (*Figure 5A*). Considering that $K_D$ for ComR/sComS interaction is in the range of 10 nM in *Streptococcus thermophilus* (*Talagas et al., 2016*), this

**Table 1.** Competence development (transformation frequency[a]) in *S. salivarius* HSISS4 derivatives

| Strains | No peptide | sComS | sBI7 |
|---|---|---|---|
| Wild-type | ND | 1.1 (±0.08) E-03 | ND |
| ΔscuR | ND | 0.7 (±0.2) E-03 | ND |
| ΔsarF | ND | 1.5 (±0.5) E-03 | ND |
| P$_{32}$-scuR | ND | NA | NA |
| P$_{32}$-sarF | ND | NA | NA |

[a]calculated as the ratio of transformants (chloramphenicol-resistant CFU) to the total CFU count per 0.1 ug of linear DNA. Transformation frequencies are expressed as the arithemtic mean of three independent experiments. Geometric means ± standard deviations are provided. ND: not detected (<1.0 E-08), NA: not applicable.
DOI: https://doi.org/10.7554/eLife.47139.012

indicates that ScuR strongly binds sBI7. In contrast, the SarF titration curve barely reached a plateau at 1 μM, highlighting a much weaker affinity (*Figure 5A*). As a selectivity control, we performed the same experiment with the non-cognate fluorescent probes, and showed that neither ScuR nor SarF was capable of binding sComS (*Figure 5B*), while sBI7 was surprisingly able to bind ComR, even if the affinity is about three times weaker compared to sComS (*Figure 5A*). Considering that sBI7 is not able to activate ComR in vivo, we suspected that this interaction is ineffective (see next section).

## Selective recognition of targeted promoters

Next, we carried out in vitro mobility shift assays to assess the direct interaction between proteins and promoter probes in absence or presence of decreasing concentration of synthetic peptides. We included a P$_{comX}$ probe as a negative control for ScuR/SarF. We were again able to corroborate our promoter activity data at the magnitude and protein-peptide/DNA specificity levels (*Figure 6A* and *Figure 6—figure supplement 1A, B and C*). While insensitive to sComS (*Figure 6—figure supplement 1A*), the ScuR regulator displayed the weakest affinity for P$_{comS}$, and the strongest one toward P$_{sptA}$, starting from 0.08 μM of sBI7 peptide and complexing the total amount of probe at maximal

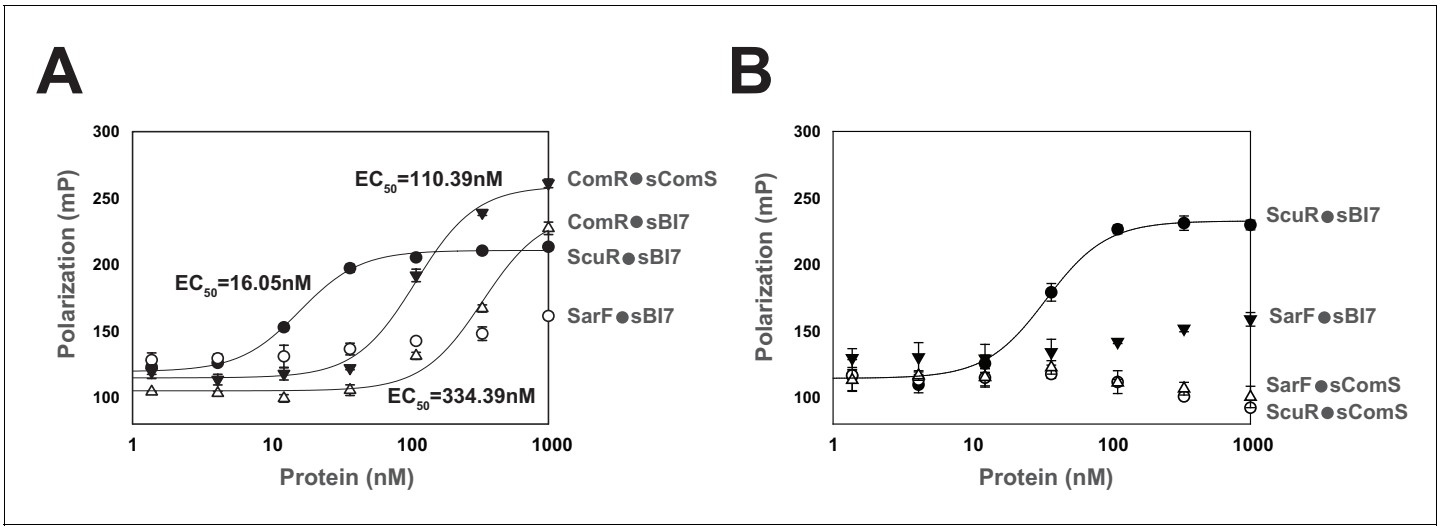

**Figure 5.** Peptide binding specificity toward ComR paralogs. (**A and B**) Fluorescence polarization of synthetic peptide-regulator pairs. Effective concentration of peptide was 10 nM. Hill equation was used to fit sigmoid curves on profiles that reach saturation, namely the ComR•sComS, ComR•sBI7, and ScuR•sBI7 couples, and calculate an EC$_{50}$ affinity factor. Experiments (**A**) and (**B**) were performed independently. (**A**) Titration of ComR, ScuR and SarF with a fixed concentration of their cognate or non-cognate peptide. (**B**) Titration of ScuR (circle) and SarF (triangles) with a fixed concentration of their cognate (sBI7; black symbols) or non-cognate (sComS; open symbols) peptides. Experimental values represent averages (with standard error of the mean, SEM) of at least three independent replicates. Some standard errors are too small to be visualized.
DOI: https://doi.org/10.7554/eLife.47139.013

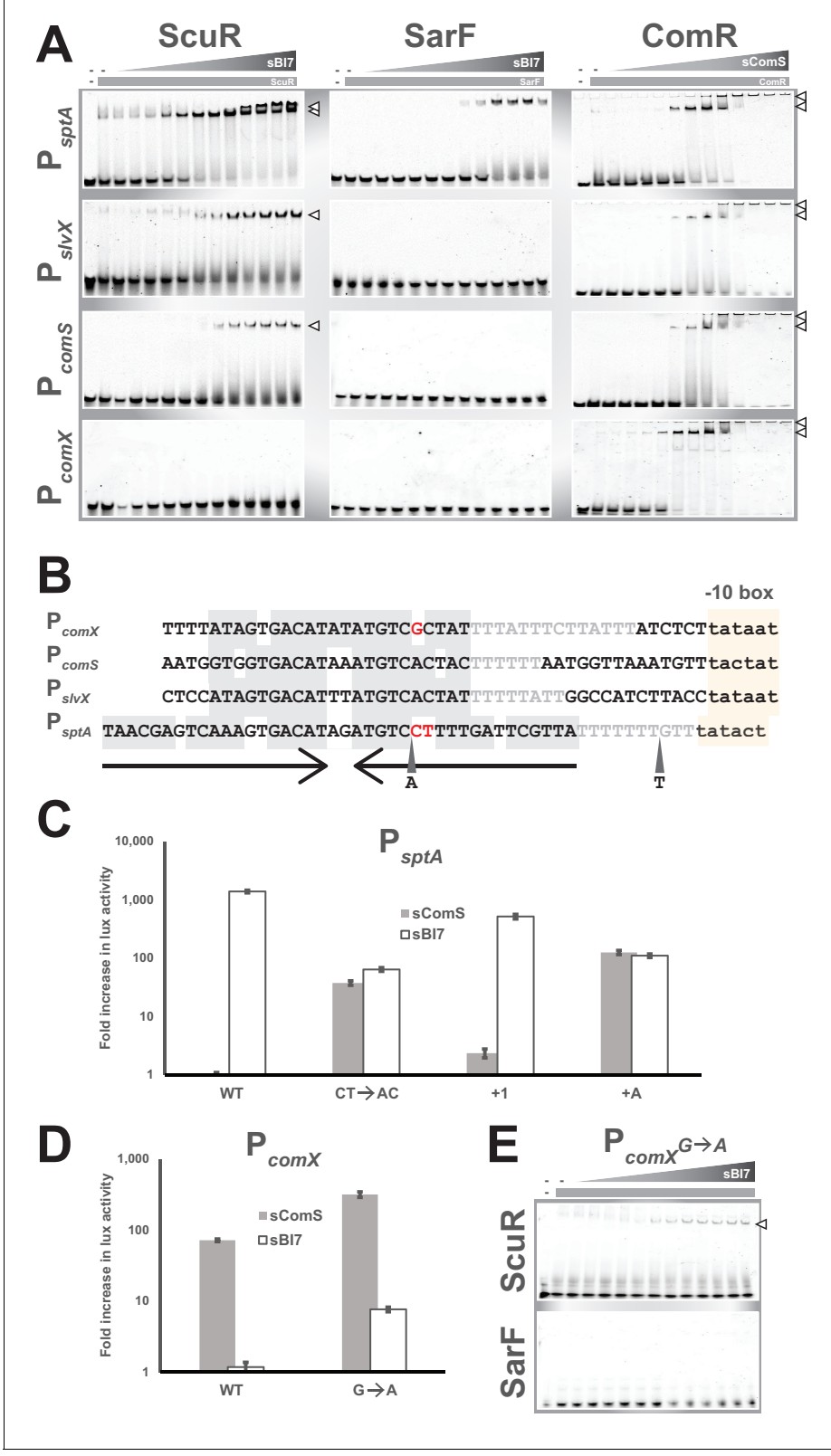

**Figure 6.** Singularities in promoter recognition of ComR paralogs. (**A**) Mobility shift assays of *comX*, *comS*, *slvX* and *sptA* promoter probes conducted with purified ComR paralogs and decreasing concentrations of their cognate peptide (gray triangles; 2:2 dilutions from 20 µM). Probes are 30 bp (or 40 bp for P*sptA*), were Cy3-conjugated and used at 40 ng. Protein concentration remained constant (gray boxes; 4 µM). Open triangles show ternary complexes (peptide-regulator-DNA) (***Figure 6—figure supplement 1***). (**B**) Nucleotide alignment of *comX*, *comS*, *slvX* and *sptA* promoters. The
*Figure 6 continued on next page*

*Figure 6 continued*

(pseudo-)palindromic stretches (converging arrows) and the sigma-bound DNA sequence (−10 boxes) are highlighted in gray or yellow, respectively. The characteristic T-rich region is gray-font. Red-marked nucleotides highlight the potential mismatches in the pseudo-palindromic structure of P$_{comX}$ and P$_{sptA}$ that were substituted to restore a genuine dyad symmetry sequence (see *Figures 6C* and *4D*). A and T represent the position and nature of single nucleotide insertion in the *sptA* promoter (see *Figure 6C*) (*Figure 6—figure supplement 1*). (C and D) Fold increase in maximal luciferase activity of WT and mutated promoters of *sptA* (C) or *comX* (D) exposed to sBI7 or sComS (1 μM). Nucleotides substitutions and insertions are shown in *Figure 6B*. Experimental values represent the averages (with standard error of the mean, SEM) of at least three independent biological replicates. (E) Mobility shift assays of mutated *comX* promoter probes conducted with a single concentration of ScuR or SarF (gray boxes; 4 μM) and decreasing concentrations of sBI7 peptide (gray triangles; 2:2 dilutions from 20 μM). Open triangles showcase ternary complexes (peptide-regulator-DNA).

DOI: https://doi.org/10.7554/eLife.47139.014

The following figure supplement is available for figure 6:

**Figure supplement 1.** Promoter and peptide specificities toward ScuR and SarF.

DOI: https://doi.org/10.7554/eLife.47139.015

concentrations (presence of a doublet presumably due to a second state of oligomerization or a conformational change) (*Figure 6A*). We observed similar results with a single concentration of sBI7 and decreasing concentrations of ScuR on P$_{sptA}$ and P$_{slvX}$ probes (*Figure 6—figure supplement 1B*). We also reconciled the above-mentioned discrepancy between luciferase and fluorescent polarization assays. Indeed, we showed that even if ComR is able to bind the non-cognate peptide sBI7 (*Figure 5A*), this regulator-pheromone pair was unable to generate a ternary complex with DNA (*Figure 6—figure supplement 1C*). Moreover, in comparison to the sComS-bound ComR, the ScuR•sBI7 affinity for P$_{comS}$ and P$_{slvX}$ appeared weaker with a less stable complex (probe smear) (*Figure 6A*). Consistently, the SarF•sBI7 complex specifically bound P$_{sptA}$, even if with a lower affinity compared to the ScuR•sBI7 pair. Remarkably, the ScuR and SarF binding linearly increased with the amount of peptide, contrasting with ComR, which showed a smaller interval between sub-activating and saturating concentrations of sComS. We cannot rule out that the ScuR/SarF native peptide would have different activator properties compared to sBI7. However, this suggests ScuR and SarF have a different dynamic of binding compared to ComR, which might reflect the congruence between reactivity and physiological function. Finally, the ComR•sComS pair could unexpectedly occupy the P$_{sptA}$ probe. Apparently, the variation in promoter primary sequence (specifically in the palindrome stretch) compared to P$_{comS}$ or P$_{slvX}$ is not sufficient to prevent ComR binding (*Figure 6B*). This indicates that the *sptA* promoter topology might be crucial to dictate the specific ScuR/SarF-driven transactivation. Indeed, a 40bp-long oligonucleotide could be sufficient for recognition by the HTH domain of ComR, but the genomic context, such as the position of the palindrome in regard to the −10 box (as described in the next paragraph), could impair the interaction between ComR and the RNA polymerase subunits, preventing trans-activation.

The topology of ScuR, SarF or ComR responsive promoters is somewhat similar (*Figure 6B* and *Figure 6—figure supplement 1D*). The architecture of every promoter includes a conserved nucleotide stretch of dyad symmetry and a T-rich spacer that separates it from the sigma-binding −10 box (same length in all promoters, P$_{sptA}$ apart). However, the core pseudo-palindromic region of the ComR-specific P$_{comX}$ includes a mismatch, while the equivalent stretch in the ScuR- and SarF-specific P$_{sptA}$ is more extended, comprises three mismatches, and is closer to the −10 box of one nucleotide (*Figure 6B*). Considering the high degree of similarity between the promoters at the primary sequence level (*Figure 6B* and *Figure 6—figure supplement 1D*), we wanted to identify the nucleotides responsible for protein-DNA selectivity. We therefore mutated *comX* and *sptA* promoters to sensitize them toward ScuR and ComR, respectively. In P$_{comX}$, we substituted a guanosine for an adenosine (P$_{comX}^{G→A}$) to reconstitute the palindromic region observed in P$_{comS}$ and P$_{slvX}$ (*Figure 6B*). With P$_{sptA}$, we performed three kinds of mutation (*Figure 6B*). We reconstituted the symmetric region with two substitutions (P$_{sptA}^{CT→AC}$), we inserted one nucleotide in the T-rich stretch to restore the same distance between palindrome center and −10 box (P$_{sptA}^{+1}$), and finally, with the mere insertion of a deoxyadenosine in the palindrome (P$_{sptA}^{+A}$), we redesigned both space and palindrome. Consistently, all these mutations impacted on selectivity to a different extent, rendering the engineered *sptA* promoters sensitive to sComS and sBI7, presumably through activation of ComR and ScuR/SarF, respectively (*Figure 6C and D*). In agreement with this, ScuR, but not SarF, was able to shift the P$_{comX}^{G→A}$ probe in presence of sBI7 (*Figure 6E*). As competence entry hinders

cell fitness (*Haijema et al., 2001*; *Mignolet et al., 2018*; *Nester and Stocker, 1963*; *Zaccaria et al., 2016*) and as the mutation in *comX* promoter enhanced the ComRS-mediated activation (*Figure 6D*), we suspect that evolution maintains a selective pressure to ensure an appropriate expression of *comX* in time and scale, compatible with the bacterial life cycle.

## Pheromone-induced ScuR promotes bacteriocin production

The lower reactivity of salivaricin promoters toward ScuR (vs ComR) pheromone prompted us to investigate the phenotypical output at the bacteriocin production level. Hence, we performed a standard bacteriocin test on soft overlay and showed that the *scuR*++ (but not *sarF*++) overexpression mutant is able to produce an inhibition halo in bacteriocin tests (*Figure 7A*). This phenotype depends on ComA, the salivaricin secretion system (*Mignolet et al., 2018*), but not SptA. Even if the overexpression of *scuR* is more potent, sBI7 induced a small halo formed by inhibition around the WT strain for concentrations ranging from 1 nM to 1 μM (*Figure 7B* and *Figure 7—figure*

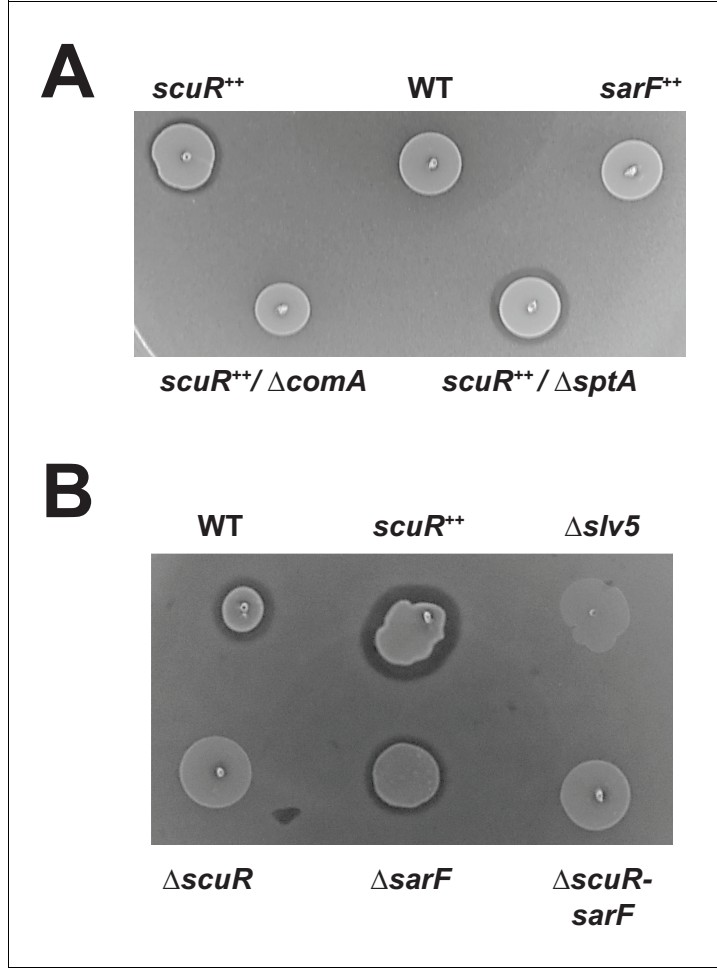

**Figure 7.** Activated ScuR drives bacteriocin production. (**A and B**) Bacteriocin inhibition assay of *S. salivarius* WT and mutant derivatives. The indicator strain (*L. lactis*) was embedded in the top soft agar layer, while sBI7 was added into the bottom agar layer as required. Producer strains were spotted on top of the two agar layers. (**A**) Killing properties of *scuR* or *sarF* overexpression mutants compared WT without sBI7 induction. (**B**) Effect of sBI7 addition (1 μM) on WT strain and various *scuR/sarF* mutants. *scuR*++ and bacteriocin null (Δ*slv5*) mutants were used as positive and negative controls, respectively (*Figure 7—figure supplement 1*).
DOI: https://doi.org/10.7554/eLife.47139.016

The following figure supplement is available for figure 7:

**Figure supplement 1.** Bacteriocin production through ScuR activation with sBI7.
DOI: https://doi.org/10.7554/eLife.47139.017

*supplement 1A*). This effect was ComR-independent (*Figure 7—figure supplement 1B*) and absent in single Δ*scuR* or double Δ*scuR-sarF* mutants, or in a bacteriocin-deficient strain (Δ*slv5*), demonstrating that the toxicity was due to bacteriocins and mediated by ScuR (*Figure 7B*).

## Discussion

When we unveiled the interplay between competence and predation in *S. salivarius* (*Mignolet et al., 2018*), we were intrigued by the atypical coupling through the single regulator ComR. Although experimental data predict that bacteriocin secretion precedes competence at low ComS concentration, the two functions are intricately associated and leave a narrow window for bacteriocin-proficient non-competent state. Therefore, the presence of a second pheromone sensor with its own selectivity would theoretically confer more flexibility to probe the environment and survey kin physiology. At least, it would duplicate the number of inputs capable of inducing the predation response. As suggested by our current and previous DNA-binding assays (*Figure 6A*) (*Talagas et al., 2016*), the dynamic of ComR switch state might be more robust compared to a gradual activation of ScuR. Presumably, this could be due to evolutionary constrains on *S. salivarius* ComR that masters dual functions with collateral toxicity, therefore requiring a fine-tuned activation. Hence, ScuR might be more permissive to environmental peptides or to non-kin pheromones and favor interspecies crosstalk. In this perspective, the fact that no coding sequence in the *S. salivarius* genome matches the sequence constrains observed for our peptides (see Materials and methods) prompts us to speculate that the pheromone could be secreted by an alien organism. In case of friends, such cues could be used to coordinate their behavior, while signals from enemies would be processed to mount an attack. Alternatively, natural ScuR overexpression could self-activate the system without any pheromone requirement, though this hypothesis contrasts with the current knowledge on RRNPP regulatory mechanisms.

Except for seven strains belonging to *Streptococcus pneumoniae*, *Streptococcus macacae*, *Streptococcus equinus*, *Streptococcus agalactiae*, *Streptococcus ferus* and *Streptococcus pantholopis* species that possess at least one homolog for the four genes of the ScuR system (with a similar genomic organization), ScuR or SarF are exclusively spread in *S. salivarius* strains (at least 31 sequenced strains) (*Supplementary file 5*). ScuR/SarF and BlpRH usually do not co-exist in the same strain of *S. salivarius* (*Figure 8A* and *Supplementary file 5*), suggesting a progressive exclusion of one system by the other one. Curiously, the direct control of competence and bacteriocin network seems to drift from a full TCS control (ComCDE and BlpRHC) in *S. pneumoniae* to a full Rgg regulation (ComRS and ScuR) in most of *S. salivarius* strains, via a hybrid mechanism (ComRS and BlpRHC) in mutans, bovis and pyogenes streptococci groups (*Figure 8B*) (*Shanker and Federle, 2017*). Why opposing extracellular vs intracellular sensing mechanisms that occur in streptococci remains a puzzling question. As a member of the gastro-intestinal tract (*Delorme et al., 2015*; *Van den Bogert et al., 2014*), *S. salivarius* is under a considerable selective pressure, competing for resources and territories in a constantly changing environment. The most likely hypothesis is that a cytoplasmic receptor could be better protected from communication interferences. Indeed, quenching molecules extruded by competitors encounter the chemically selective semi-permeable cell membrane to penetrate into the cell. A second hypothesis is that internal cues (due to cell physiology, alarmone or cytokinesis) might impinge on the nutritional oligopeptide transport system (Opp) to globally modulate small peptide inward fluxes and make the cell transiently communication-less or superreactive to pheromones in particular stressful situations.

All our phenotypical and molecular data converge to demonstrate that ScuR is strictly dedicated to predation in contrast to competence. Although highly similar to P$_{comS}$ and P$_{slvX}$ at primary sequence level, the *comX* promoter can be neither occupied (*Figure 6A*) nor activated (*Figures 2A* and *6D*) by ScuR. However, this regulator turns on P$_{comS}$ and should somehow modulate the ComRS activity through a weak momentum boost on the positive feedback loop. An obvious reason for this discrepancy might be that the ScuR-driven expression of *comS* (*Figures 2A* and *4D*) is weaker compared to our previous observations for ComR (*Mignolet et al., 2018*). Therefore, it is not sufficient to reach the activation threshold, but a basal production of the ComA transporter encoded in operon with *comS* might be required to extrude bacteriocins. Alternatively, we suspect that promoters are not responsive to ComR and ScuR pheromone within the same timeframe. In this case,

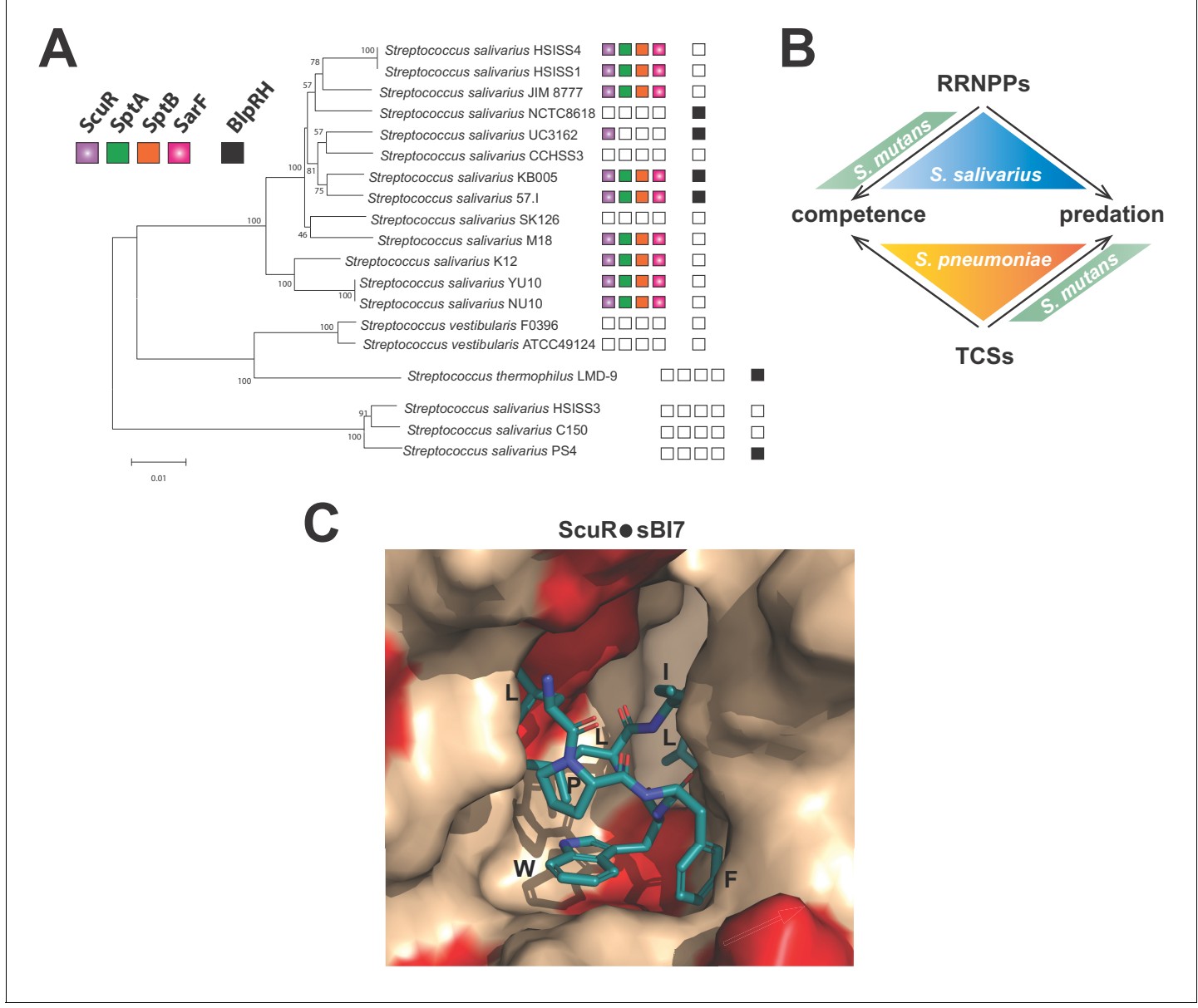

**Figure 8.** Distribution and activation of ComR paralogs for predation control in *S. salivarius* (**A**) Conservation of ScuR-encoding locus, and BlpRH system across *S. salivarius* (***Supplementary file 5***). Functional BlpRH pair (black), ScuR (purple), SptA (green), SptB (orange), and SarF (pink) were screened for homologs in various *S. salivarius* strains. The phylogenetic tree (100 bootstrap replicates) was adapted from ***Yu et al. (2015)***. An empty box means that no functional ortholog was found in the strain genome. Scale bar: 0.01 substitutions per site. (**B**) Figurative illustration of RRNPPs vs two-component systems (TCSs) drift toward competence and predation regulation in model streptococci (*S. pneumoniae*, *S. mutans* and *S. salivarius*). Arrows symbolize the species-specific control of quorum sensing regulators on developmental processes. (**C**) Close view of the ScuR peptide-binding pocket. The model of the peptide-bound form of ScuR was obtained by homology modeling using the i-TASSER server and the ComR•ComS•DNA complex (PDB ID 5JUB) as template. It is shown as surface colored in beige except for residues substituted in SarF, wich are highlighted in red. The bound sBI7 peptide is shown in blue sticks and its residues LPFWLIL are labeled. The C-terminal glycine, deeply buried in the pocket, is not visible in this view. The figure was prepared using the graphic software PyMol.

DOI: https://doi.org/10.7554/eLife.47139.018

ScuR might promote ComS production in a time interval during which the *comX* promoter is silenced or unreactive.

As for SarF, it would exclusively control SptA/B production. According to our overexpression phenotype in a *scuR* deletion mutant (***Figure 2D***) and the peptide-promoted occupancy of P*sptA*

(*Figure 6A*), the regulation turns out to be primarily direct. However, SarF might indirectly alter P$_{sptA}$ responsiveness through an inhibitory effect on ScuR activity, given that *sarF* loss provokes a faint over-activation of ScuR (*Figure 4D*).

Considering their degree of similarity, ComR, ScuR and SarF feature a pronounced selectivity in their pheromone nature, their DNA occupancy and their transcriptional activation rate. Rational alterations as small as single point mutations or one-nucleotide insertion (*Figure 6C, D and E*) reshuffle or partially switch the permissiveness of the tripartite complex (pheromone-regulator-promoter). In line with our observations about the higher affinity of sBI7 toward ScuR vs SarF (*Figure 5A*), in silico 3D-modelling of the sBI7 binding mode on ScuR (derived from the *S. thermophilus* ComR structural scaffold) (*Figure 8C*) together with sequence comparison with SarF (*Figure 1—figure supplement 1B*) suggests that, among residues directly involved in peptide binding, the I252M substitution in the α-helix 14 would most probably results in a clash between the conserved sBI7 tryptophan and the SarF residue M252. This emphasizes that co-evolving (cytoplasmic) receptors robustly compartmentalize their inputs and outputs, even if they fulfill redundant or complementary functions.

The screening strategy we applied during this study appears productive and robust. Presumably, we identified artificial peptides with the greatest affinity or activation capacity toward their sensors, while the cognate pheromone(s) is still under investigations. Nevertheless, this approach is highly relevant in a context where the actual ligands are often not detected by genome analysis or standard lab methods. With a key and specific target promoter, it could be applicable beyond the ComR paralog scope, typically for other members of the RRNPP family or even other types of intracellular receptors. Furthermore, small adaptation (translocation of the encoded peptides via a signal sequence) of this screen could be applied to survey small extracellular peptide sensors such as membrane-spanning histidine kinases. At the biological level, the identification of inducing peptides, even non-native, will be an important step to better understand the function and the activation mode of these regulatory proteins. In addition, such a screen would serve to optimize the efficacy/reactivity of a signaling pathway for biotechnological applications.

To conclude, the discovery of an alternative predation signaling network, prevalent in *S. salivarius* species (*Figure 8A*), which maintains competence in an off state would be beneficial in the scope of probiotics and human health. Indeed, we might contemplate taking advantage of the endogenous populations of commensal *S. salivarius* and provide optimized pheromones (e.g. in food, pill, ointment) to mobilize bacteriocins (*Hols et al., 2019*). By minimizing genetic drifts of the producer cells, this could enhance inter-species competition and locally clear external epithelia (e.g. mouth, intestine, skin, and vagina) from specific pathogens or hamper their settlement/growth inside the niche.

## Materials and methods

### Strains and growth conditions

*Streptococcus salivarius* HSISS4 and derivatives were grown at 37°C without shaking in M17 (Difco Laboratories, Detroit, MI) or in CDM (*Fontaine et al., 2013*) supplemented with 1% (w/v) glucose (M17G, CDMG, respectively). *Escherichia coli* TOP10 (Invitrogen) were cultivated with shaking at 37°C in LB. Electrotransformation of *E. coli* was performed as previously described (*Mignolet et al., 2016*). *Lactococcus lactis* was grown in M17 broth with 1% glucose at 30°C without shaking. Agar 1.5% (w/v) was added into M17 and LB plates, and bacteriocin inhibition tests were assayed on M17 plates containing 0.3% agar. D-xylose (0.1 or 1%; w/v), ampicillin (250 µg.ml$^{-1}$), spectinomycin (200 µg.ml$^{-1}$), chloramphenicol (5 µg.ml$^{-1}$; except if otherwise stated), erythromycin (10 µg.ml$^{-1}$), or 5-FOA (1 mg.ml$^{-1}$) (Melford Laboratories) were added as required. Synthetic peptides and sComS (purity of 95%; 1 µM, except if otherwise stated) were supplied by Peptide2.0 Inc (Chantilly, VA, USA) and resuspended in DMSO. Solid plates inoculated with *S. salivarius* cells were incubated anaerobically (BBL GasPak systems, Becton Dickinson, Franklin lakes, NJ) at 37°C.

### Bacterial strains, plasmids, oligonucleotides, and growth conditions

Bacterial strains, plasmids, oligonucleotides, and growth conditions used in this study are listed and described in *Supplementary file 1*.

## Competence induction, transformation rate and engineering of mutants

To induce competence, overnight CDMG precultures were diluted at a final $OD_{600}$ of 0.05 in 300 µl (10 ml concerning the randomized peptide screen) of fresh CDMG and incubated 75 min at 37°C. Then, the pheromone sComS was added as well as DNA (overlapping PCRs or plasmids) and cells were incubated for 3 hr at 37°C before plating on M17G agar supplemented with antibiotics where required. For transformation assays, a *cat*-borne linear PCR product (0.1 µg) used to delete a 'neutral' gene, *HSISS4_00145*, was added to 300 µl culture samples supplemented with sComS. Cells were plated on antibiotic-supplemented and –free medium. The transformation frequency was calculated as the number of chloramphenicol-resistant CFUs per ml divided by the total number of viable CFUs per ml. Null-mutants were constructed by exchanging (double homologous recombination) the coding sequences (CDS) of target genes (sequence between start and stop codons) for either chloramphenicol or erythromycin resistance cassette. If stated, mutants were cleaned for the *lox* site-flanked resistance cassette, as previously described (*Fontaine et al., 2010*). In case of deletion of multiple CDSs, the region between the start codon of the first CDS and the stop codon of the last CDS was deleted. Integration of the antibiotic resistance cassette at the right location was subsequently checked by PCR. The promoter of the *sptA* gene was fused to the *luxAB* reporter genes and inserted with a chloramphenicol resistance cassette at the permissive tRNA threonine locus (*HSISS4_r00061*) by double homologous recombination. In case of Δ*scuR* and Δ*sarF* in-frame deletion, we used the two-step selection/counter-selection strategy previously described (*Mignolet et al., 2018*). We transformed the wild-type strain with an overlapping PCR product composed of 4 fragments: (I) the upstream region of *scuR* or *sarF* genes, (II) the downstream region of *scuR* or *sarF* genes, (III) a cassette that includes the erythromycin resistance gene (*erm*) and a gene encoding the orotate transporter *oroP*, and finally (IV) the downstream region of *scuR* or *sarF* genes. We selected a first event of double recombination on medium supplemented with erythromycin. Next, we selected an intramolecular recombination between region (I) and (IV) that excises the *erm-oroP* cassette by growing cells on M17G supplemented with the toxic 5-fluoro-orotic acid (5-FOA) compound. In absence of *oroP*, 5-FOA is not able to cross the membrane and penetrate the cytoplasm where it is deleteriously incorporated in the nucleotide metabolic pathway (*Overkamp et al., 2013*). At final, we engineered an in-frame deletion mutant of *scuR* or *sarF* in which the first seven codons were fused to the last six codons without any cassette scar.

## ComR, ScuR and SarF purification

The PCR-amplified *scuR-StrepTag* and *sarF-StrepTag* genes were cloned into the pBAD-*comR-ST* vector. The ComR-StrepTag, ScuR-StrepTag and SarF-StrepTag recombinant proteins were overproduced *in E. coli* and purified as previously described (*Fontaine et al., 2013*) in standard native conditions on Strep-Tactin agarose beads (IBA).

## Randomized peptide screen

We decided to express peptides derived from a ComS backbone, reasoning that ComR, ScuR and SarF could accommodate similar peptides. To prevent export, we discarded the N-terminus and empirically selected the last 12 residues of ComS that still sustains ComR activation. Practically, we only randomized the last seven residues (~1.3 billion combinations) to keep the library representativeness compatible with the transformation rate of *S. salivarius*. To generate the two DNA libraries encoding randomized sequence of small peptides, we performed overlapping PCRs to graft fragments encompassing the follow features: (1) a 5' recombination arm (for the ectopic tRNA^ser locus), (2) the *xylR* gene that codes for the xylose responsive regulator, (3) either $P_{xyl1}$ (library I) or $P_{xyl2}$ (library II) translationally-fused to a 12 codons-long gene for which the last seven are randomized, (4) the *specR* gene, and (5) a 3' recombination arm (for the ectopic tRNA^ser locus). To obtain the randomized DNA stretch, we used a 78 nucleotides-long primer degenerated at 21 contiguous positions. Next, we transformed these two libraries into strains containing the *sptA* promoter translationally-fused to the *cat* gene (chloramphenicol resistance) in which the associated SpecR gene was excised by the previously described cre-lox method (*Fontaine et al., 2010*). The initial backgrounds of these strains were either a *comR*-overexpressing ($P_{xyl1}$-*comR*) or a salivaricin-deprived (Δ*slv5*) strain. We plated transformed cells on solid medium supplemented with xylose (either 0.1 or 1%), chloramphenicol (2 mg.ml$^{-1}$) and spectinomycin (200 mg.ml$^{-1}$) and incubated

overnight. We re-streaked single colonies on fresh chloramphenicol and spectinomycin solid medium supplemented or not with xylose. We finally collected clones that displayed an increased in growth on xylose vs non-xylose medium (except for the clone BM1 that we used as a negative control).

### Genome scanning for native peptide identification

To seek for the cognate peptide of ScuR/SarF in *S. salivarius* HSISS4 genome, we extract all open reading frames (ORF; start codon = AUG, GUG, TUG or CUG) of more than eight codons from HSISS4 chromosome with the CLC Main Workbench 7.0 software (https://www.qiagenbioinformatics.com/products/clc-main-workbench/). Next, we use the motif scanner (FIMO tool) of the MEME suite (http://meme-suite.org/) to blast a peptide sequence matrix from the last seven residues of the twenty-two non redundant synthetic peptides against 63.306 ORFs. We analyzed the first 206 top hit ($p$-value threshold of $10^{-4}$) and looked for stretch positioning in the global polypeptide/protein (close to N- or C-terminus), and genetic regulatory elements (e.g. promoter, RBS and terminators) in the vicinity of the corresponding ORF. We did not identify relevant ORF/peptide.

### Fluorescence polarization (FP)

We incubated the peptide-protein complex in black well 96-well plates (Greiner, Alphen a/d Rijn, The Netherlands) and we performed anisotropic measurement with a multi-wells plate reader (Hidex Sense, Hidex, Turku, Finland) in polarization mode. Filter settings were 485/10 nm and 535/20 nm for excitation and emission, respectively (25 flashes, Lamp Power of 50, focus 5.5 mm). The octamer synthetic peptides were conjugated to FITC, an aminohexanoic acid spacer (Ahx) and an isoleucine residue linker at the N-terminus (Peptide 2.0, Chantilly, VA, USA). They were added at a final concentration of 10 nM. The purified Strep-tagged proteins were added in 3:3 serial dilutions from 1 µM in a final volume of 100 µl of binding buffer (20 mM Tris pH 7.5, 150 mM NaCl, 1 mM EDTA, 10% glycerol) and incubated 10 min at 30°C before data acquisition. Hill equation (Hill coefficient fixed at 2) was used to generate fitting curves and determine the $EC_{50}$ affinity factor (protein concentration for a half maximum response).

### Mobility shift assays (EMSA)

All double-stranded DNA fragments (30 or 40 bp) were obtained from annealing of single-stranded Cy3-labelled (at 5' end) and unlabeled oligonucleotides. Primers used are listed in the *Supplementary file 1*. Typically, a gel shift reaction (20 µl) was performed in a binding buffer (20 mM Tris-HCl pH 8.0, 150 mM NaCl, 1 mM EDTA, 1 mM DTT, 10% glycerol, 1 mg.ml$^{-1}$ BSA) and contained 40 ng labeled probe and 4 µM (or 2:2 serial dilutions from 8 µM, if stated) of purified Strep-Tagged ScuR, SarF or ComR. When necessary, sBI7 or sComS peptides were added (either 2:2 serial dilutions from 20 µM, or 1 µM). The reaction was incubated at 37°C for 10 min prior to loading of the samples on a native 4–20% gradient gel (iD PAGE Gel; Eurogentec). The gel was run for 30 min at 70 V and then run at 50 V for approximately 2 hr in MOPS buffer (Tris-base 50 mM pH 7.7, MOPS 50 mM, EDTA 1 mM). DNA complexes were detected by fluorescence on the Ettan DIGE Imager with bandpass excitation filters (nm): 540/25 (Cy3) or 635/30 (Cy5) and bandpass emission filters: 595/25 (Cy3) or 680/30 (Cy5) (GE Healthcare, Waukesha, WI).

### Bacteriocin detection assay

The spot-on lawn (multilayer) detection method was performed as followed: 10 µl of overnight cultures of producer strains were diluted in fresh M17 medium (Difco Laboratories, Detroit, MI) supplemented with glucose (M17G) medium and grown to mid-log phase ($OD_{600}$ = ~0.5). In parallel, plates were casted with a bottom feeding layer (M17G 1.5% agar) supplemented with a synthetic peptide as required. Next, we mixed 100 µl of an overnight culture of *Lactococcus lactis* IL1403 (indicator strain) in pre-warmed soft M17G medium (0.3% agar) and casted it as a top layer. Finally, we spotted 3 µl of the producer strains on the top layer. Plates were incubated overnight before analysis of the inhibition zones surrounding the producer colonies.

### Measurements of growth and luciferase activity

Overnight precultures were diluted to a final $OD_{600}$ of 0.05. A volume of 300 µl of culture samples was incubated in the wells of a sterile covered white microplate with a transparent bottom (Greiner,

Alphen a/d Rijn, The Netherlands) and supplemented with synthetic peptides (1 μM, except if otherwise stated) or DMSO, and xylose as required. Growth ($OD_{600}$) and luciferase (Lux) activity (expressed in relative light units) were monitored at 10 min intervals during 24 hr in a multi-well plate reader (Hidex Sense, Hidex, Turku, Finland) as previously described (*Fontaine et al., 2013*).

### Deep sequencing (RNAseq) and data processing

*S. salivarius* WT, Δ*scuR*, Δ*sarF*, *scuR* [++] or *sarF* [++] strains were pre-cultured overnight in CDMG at 37° C. They were resuspended in 50 ml of fresh pre-warmed CDMG to a final $OD_{600}$ of 0.05 and grown for approximately 2 hr 30 min ($OD_{600}$ = 0.3) at 37°C. Cells were harvested by centrifugation (10 min; $4,050 \times g$), the supernatants were discarded and the cell pellets were frozen with liquid nitrogen. Finally, RNA was extracted using the RiboPure bacteria kit (Ambion-Life Technologies) and the protocol provided by the manufacturer, with protocol changes to cell lysis and RNA precipitation. For lysis, cells were resuspended in RNAwiz buffer (Ambion-Life Technologies) supplemented with Zirconia beads and shaked for 40 s (four times) in a fastPrep homogenizer device (MP biomedicals). For RNA precipitation, a 1.25-ethanol volume (instead of 0.5) was added to partially purified RNAs. Total RNA was checked for quality on a RNA Nano chip (Agilent technologies) and concentration was measured using Ribogreen assay (Life technologies). rRNA depletion was performed on 2 μg total RNA with the Ribo-Zero rRNA removal kit for Gram-positive bacteria (Illumina) according to manufacturer's instructions. Total stranded mRNA libraries were prepped with the NEBNext Ultra Directional RNA Library Prep kit for Illumina (New England Biolabs). Library PCR was executed for 15 cycles. Quality of the libraries was evaluated with the use of a High sensitivity DNA chip (Agilent technologies) and concentrations were determined through qPCR according to Illumina protocol. Libraries were sequenced on a NextSeq 500 high-throughput run with 76 bp single reads. 2.3 pM of the library was loaded on the flowcell with a Phix spike-in of 5%. Sequenced mRNAs generated several million reads that were mapped on the WT *S. salivarius* chromosome and processed with both bowties V0.12.9 (http://bowtie-bio.sourceforge.net/bowtie2) and samtools V0.1.18 (http://samtools.sourceforge.net/) algorithms to yield BAM files containing the read coordinates. We imported these files into SeqMonk V0.23.0 (www.bioinformatics.babraham.ac.uk/projects/) to assess the total number of reads for each coding sequence (CDS). The dataset was exported into an excel file for further analyses. First, the dataset was standardized to CDS-mapped reads per million overall reads. Then, we estimated a ratio of CDS-mapped reads in mutants vs WT. All RNAseq data were deposited in the GEO database under accession number GSE120640.

### In silico modeling

Homology modeling of ScuR was performed by the i-TASSER server for protein 3D structure prediction (*Zhang, 2008*) by using the crystal structure of the ComR•sComS•DNA complex from *S. thermophilus* (PDB ID 5JUB) (*Talagas et al., 2016*) as template. The resulting model displayed a confidence score of 1.45 and a TM-score of 0.92 ± 0.06, reflecting a high confidence and a topology highly similar to the template (*Roy et al., 2010*; *Yang et al., 2015*; *Zhang, 2008*). The bound peptide was manually modified to accommodate the sequence of sBI7 and the resulting complex was optimized using the Gromacs energy minimization server NOMAD-REF (*Lindahl et al., 2006*). The figures of 3D structures were prepared using PyMol (The PyMOL Molecular Graphics System, Version 2.0 Schrödinger, LLC).

## Acknowledgements

Funding support was from FNRS and IUAP grants to TC and PH. PH is Senior Research Associate at FNRS.

## Additional information

**Competing interests**

Johann Mignolet: Dr J Mignolet is currently hired by the Syngulon company; Part of the data presented in this paper are under the authority of a patent (Application number: EP18207694.3). Pascal

Hols: Pr P Hols is member of the Syngulon scientific advisory board. The other authors declare that no competing interests exist.

## Funding

| Funder | Grant reference number | Author |
|---|---|---|
| Fonds De La Recherche Scientifique - FNRS | PDR T.0110.18-F | Pascal Hols |
| BELSPO | Project P7/28 | Tom Coenye<br>Pascal Hols |
| Walloon Region (Belgium) | RPR1620203 | Johann Mignolet |

The funders had no role in study design, data collection and interpretation, or the decision to submit the work for publication.

## Author contributions

Johann Mignolet, Conceptualization, Supervision, Validation, Investigation, Methodology, Writing—original draft, Project administration, Writing—review and editing; Guillaume Cerckel, Julien Damoczi, Validation, Investigation; Laura Ledesma-Garcia, Methodology, Writing—review and editing; Andrea Sass, Writing—review and editing; Tom Coenye, Resources, Funding acquisition, Writing—review and editing; Sylvie Nessler, Formal analysis, Methodology, Writing—review and editing; Pascal Hols, Conceptualization, Resources, Supervision, Funding acquisition, Methodology, Writing—original draft, Project administration, Writing—review and editing

## Author ORCIDs

Johann Mignolet https://orcid.org/0000-0002-3721-4307

## Decision letter and Author response

Decision letter https://doi.org/10.7554/eLife.47139.028
Author response https://doi.org/10.7554/eLife.47139.029

# Additional files

## Supplementary files

• Supplementary file 1. Bacterial strains, plasmids and oligonucleotides list, and DNA engineering.
DOI: https://doi.org/10.7554/eLife.47139.019

• Supplementary file 2. RNAseq data on WT, ΔscuR and ΔsarF strains.
DOI: https://doi.org/10.7554/eLife.47139.020

• Supplementary file 3. RNAseq data on WT, $scuR^{++}$ and $sarF^{++}$ strains.
DOI: https://doi.org/10.7554/eLife.47139.021

• Supplementary file 4. Random peptide list.
DOI: https://doi.org/10.7554/eLife.47139.022

• Supplementary file 5. Conservation of ScuR-SarF system in streptococci.
DOI: https://doi.org/10.7554/eLife.47139.023

• Transparent reporting form
DOI: https://doi.org/10.7554/eLife.47139.024

## Data availability

Sequencing data have been deposited in GEO under accession code GSE120640.

The following dataset was generated:

| Author(s) | Year | Dataset title | Dataset URL | Database and Identifier |
|---|---|---|---|---|
| Mignolet J, Cerckel G, Damoczi J, Ledesma-Garcia L, | 2018 | Molecular singularities in a pheromone sensor triumvirate desynchronize the competence- | https://www.ncbi.nlm.nih.gov/geo/query/acc.cgi?&acc=GSE120640 | NCBI Gene Expression Omnibus, GSE120640 |

| Coenye T, Hols P | predation interplay in the human commensal Streptococcus salivarius |

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
