## [Decision Letter]

Thank you for submitting your article "Subtle selectivity in a pheromone sensor triumvirate desynchronizes competence and predation in a human gut commensal" for consideration by *eLife*. Your article has been reviewed by three peer reviewers, one of whom is a member of our Board of Reviewing Editors, and the evaluation has been overseen by a Reviewing Editor and Gisela Storz as the Senior Editor. The following individuals involved in review of your submission have agreed to reveal their identity: Michael Federle (Reviewer #3).

The reviewers have discussed the reviews with one another and the Reviewing Editor has drafted this decision to help you prepare a revised submission.

Summary:

Previously, Mignolet et al. reported that in the human gut commensal *Streptococcus salivarius*, the cytoplasmic Rgg/RNPP regulator ComR couples competence to bacteriocin-mediated predation. In the current manuscript, a paralogous sensor duo, ScuR and SarF was characterized which circumvents ComR in order to disconnect competence and bacteriocin production. These three transcriptional factors show a high degree of similarity on the amino acid level. The authors developed a genetic screen to discover cognate peptide pheromones activating the sensors. By extensive mutational and biochemical analyses the systems was characterized in detail at the molecular level. The authors propose that this alternative ScuR pathway confers increased flexibility in regard to the timing and the magnitude of offensive/defensive responses. It might provide an advantage to desynchronize competence activation and secretion of antimicrobial compounds.

Essential revisions:

1) Figure 3F: A control with ComR binding or not binding sBI7 should be included. This is needed to further emphasize specificity.

2) Puzzling and at the same time very interesting is the fact that no coding sequence in the *S. salivarius* genome matches the sequence constrains observed for the identified signalling peptides. As the authors stated, such peptides could be produced by other microorganisms. You should check the data base of relevant bacteria for presence of similar peptides. According to your model, these bacteria should be gut bacteria most likely pathogens such as *Salmonella* etc. Also, supernatants of such bacteria could be tested for activation of the system.

3) The explanation why the ComR•sComS pair can unexpectedly occupy the P*_sptA_* probe was difficult to understand. We recommend elaborate this point because it questions in part the specificity.

4) The in vitro data is highly convincing, although it would benefit from additional in vivo experimentation (natural competence assays with protein/pheromone variants and monitoring expression of competence/bacteriocin genes). As additional controls, you need to monitor expression of competence genes (*ssb, comX*, etc.). Please show that ScuR/SarF deletions do not affect competence gene expression specifically in an in vivo assay.

5) Subsection “Pheromone-induced ScuR promotes bacteriocin production”: Which bacteriocins are produced? Can you add more detail in this section? The regulation of bacteriocins/predation is advertised as a major point in this work and at least in the main text, few experiments are done on this part. Possibly you should show a read out of bacteriocin expression (RT-PCR, western, or another of the author's choice) in the mutant strains as a control. A lot of that type of data and assay design seems to be in the Cell Reports paper Mignolet et al. (2018). Can that be better discussed here and used as a template for assays in this paper?

6) For the peptide docking and modeling data you need to provide the statistics of the model and docking fit provided by the programs you used.

---

## [Author Response]

Summary:Previously, Mignolet et al. reported that in the human gut commensal Streptococcus salivarius, the cytoplasmic Rgg/RNPP regulator ComR couples competence to bacteriocin-mediated predation. In the current manuscript, a paralogous sensor duo, ScuR and SarF was characterized which circumvents ComR in order to disconnect competence and bacteriocin production. These three transcriptional factors show a high degree of similarity on the amino acid level. The authors developed a genetic screen to discover cognate peptide pheromones activating the sensors. By extensive mutational and biochemical analyses the systems was characterized in detail at the molecular level. The authors propose that this alternative ScuR pathway confers increased flexibility in regard to the timing and the magnitude of offensive/defensive responses. It might provide an advantage to desynchronize competence activation and secretion of antimicrobial compounds.

We are grateful toward you as well as the reviewers for your enthusiasm and your inspiring comments about our work. We modified the manuscript in agreement with your expectations. Moreover, we inserted several supplementary experiments that we described hereunder and we reorganized figure panels to increase the amount of data in main figures.

Essential revisions:1) Figure 3F: A control with ComR binding or not binding sBI7 should be included. This is needed to further emphasize specificity.

We completely agree with reviewers and included a control curve in fluorescence polarization experiments to show (absence of) binding between ComR and sBI7 (see Figure 5A). Please note that we performed the experiment with the ScuR-sBI7, SarF-sBI7 and ComR-sComS pairs again. We showed that ComR is able to bind sBI7 with a lower affinity compared to the cognate peptide sComS. Considering that sBI7 cannot activate the ComR pathway in vivo (luciferase test), we asked whether this interaction is productive in vitro (Mobility shift). Consistently, sBI7 is unable to induce the formation of a ternary complex of ComR with the promoter probes of *comX* or *slvX* (Figure 6—figure supplement 1C). From these data, we concluded that sBI7 can be accommodated by ComR. However, the conformational changes are not sufficient to drive the dimerization or the DNA-binding domain release and the subsequent activation of the protein-peptide complex. This strengthens our model of activation selectivity.

2) Puzzling and at the same time very interesting is the fact that no coding sequence in the S. salivarius genome matches the sequence constrains observed for the identified signalling peptides. As the authors stated, such peptides could be produced by other microorganisms. You should check the data base of relevant bacteria for presence of similar peptides. According to your model, these bacteria should be gut bacteria most likely pathogens such as Salmonella etc. Also, supernatants of such bacteria could be tested for activation of the system.

The reviewers highlighted an important issue. The discovery of small peptides from genome inspection is a particularly time-consuming task with low success rate, especially when the peptides are not encoded close to the regulator gene. The main reasons are the small size of the coding sequences and the cryptic presence of regulatory elements (RBS, promoter) as well as the fact that the peptides could be generated from the processing of proteins (usually secreted proteins such as lipoproteins) or that the peptide genes could be embedded in other longer coding sequences. On the top of it, constrains on the sequence (the conserved tryptophan apart) to identify the cognate peptide are dreadfully low. In other word, the number of hits in the genome through tblastn (as described in Materials and methods section) is too high to individually test all of them. As suggested by reviewers, we interrogated the human microbiome on NCBI database with the consensus sequence LPFWLVLG as query. Unfortunately, we faced two unsolvable issues. First, we listed a huge number of hits insuperable without another cross-checking information. Second, many of small peptides are not annotated/predicted in bacterial genomes and therefore are invisible for blastp prospects (a tblastn would have produced at least 10 times more hits).

Finally, we assume that our synthetic screen based on a high selective pressure (antibiotic survival) identified peptides with the highest affinity and activation capacity. Albeit highly interesting for medical applications, these peptides might potentially be a disadvantage in nature, as they exacerbate the predation behavior, leading to their counter selection. Therefore, the sequence of the cognate peptide might be quite divergent compared to the consensus sequence.

We put a lot of effort to identify the native pheromone of the ScuR system. For instance, we tried to co-purify the cognate peptide and ScuR by doing a pull-down of ScuR in an overexpressing mutant and analyzing the co-immunoprecipitate by mass spectrometry. In parallel, we tested the cross-activation of ComS from other streptococci (e.g. *S. pyogenes, S. mutans* and *S. suis*), given that the type II and type III ComS possess a tryptophan, even if at a different position. We also collected saliva from human origin to cultivate *S. salivarius*. All of these experiments failed to identify a peptide or to trigger P*_sptA_* or bacteriocin promoter response. As suggested by reviewers, we harvested and filtrated supernatants of strains known to be sensitive to *S. salivarius* bacteriocins (see Mignolet et al., 2018), added 30µl to *S. salivarius* cultivated in CDM and, looked for P*_sptA_* activation. We did not observe any variation of promoter activity compared to mock conditions (M17G without bacteria). As *Salmonella* (and Gram-negative bacteria in general) is not sensitive to *S. salivarius* bacteriocins and *Salmonella* is not known to co-aggregate (cooperate) with *S. salivarius* in vivo, we did not include it in our experiments. For such experiments, we cannot rule out that the production of detectable quantity of cognate peptides might require specific growth conditions. For instance, it was published that natural transformation of *S. pyogenes* cannot arise upon exogenous sComS addition but requires specific biofilm conditions (Marks et al., 2014). We regret that we cannot test more species known to be present in the human microbiome with the time frame given for resubmission in *eLife*.

To conclude, we theoretically favor the hypothesis that the ScuR/SarF cognate peptides are encoded in *S. salivarius* genome, even if we were not able to identify them. Possibly, the strain HSISS4 could be a cheater that senses the pheromones of other *S. salivarius* strains. However, we additionally speculated an alternative tantalizing mechanism in which *S. salivarius* could evaluate the physiology of friends or foes and react accordingly.

3) The explanation why the ComR•sComS pair can unexpectedly occupy the P_sptA_ probe was difficult to understand. We recommend elaborate this point because it questions in part the specificity.

We modified the text according to reviewer comments to clarify this point. In substance, we presume that the genomic context is determining in this case. Indeed, the P*_sptA_* probe is only 40bp-long. Apparently, two mere mismatch in the dyad symmetry is not deleterious for ComR binding (Figure 6A). However, all our current and previous in vivo data (Figure 2A, 2C, Figure 6C and Supplementary file 1 from Mignolet et al., 2018 [genes HSISS4_01166, 67, 68 and 69]) converge to demonstrate that ComR specifically activates P*_comX_* while ScuR specifically activates P*_sptA_*. We suspect that this discrepancy between in vivo and in vitro data could be due to the position of the -10 box in regard to the center of the palindrome: 31 nucleotides for bacteriocin, *comS* and *comX* promoters, 30 nucleotides for P*_sptA_*. This difference of 1 nucleotide could misposition (3D shift of about 35°) the dimeric ComRS complex for interaction with one subunit of the RNA polymerase, preventing an efficient transactivation.

To conclude, the ComRS system is totally ineffective to activate P*_sptA_* in vivo. These results demonstrate the biological selectivity of activation between ComR and ScuR/SarF.

4) The in vitro data is highly convincing, although it would benefit from additional in vivo experimentation (natural competence assays with protein/pheromone variants and monitoring expression of competence/bacteriocin genes). As additional controls, you need to monitor expression of competence genes (ssb, comX, etc.). Please show that ScuR/SarF deletions do not affect competence gene expression specifically in an in vivo assay.

We performed a series of additional experiments to prove that ScuR/SarF does not trigger competence nor natural transformation. First, we engineered a bunch of new luciferase reporter strains to evaluate the expression of *comX* and ComX-induced genes (namely, *ssbA, comEA, comFA, cbpD1*, and *dprA*) under sComS or sBI7 treatment (Figure 4E). We showed that, while the promoter of all these genes are turned on with sComS, they are totally insensitive to sBI7. Second, we showed that deletion of *scuR, sarF* or both genes does not affect the sComS-mediated activation of P*_comX_* (Figure 4—figure supplement 3). Finally, we assessed the transformation rate in *scuR/sarF* variants or upon sBI7 addition (Table 1). We showed that neither sBI7 addition, nor *scuR/sarF* overexpression are able to generate transformants, while the deletion of *scuR* or *sarF* has no major impact on the transformation rate. These results highlight that ScuR and SarF are not require for competence and natural transformation.

5) Subsection “Pheromone-induced ScuR promotes bacteriocin production”: Which bacteriocins are produced? Can you add more detail in this section? The regulation of bacteriocins/predation is advertised as a major point in this work and at least in the main text, few experiments are done on this part. Possibly you should show a read out of bacteriocin expression (RT-PCR, western, or another of the author's choice) in the mutant strains as a control. A lot of that type of data and assay design seems to be in the Cell Reports paper Mignolet et al. (2018). Can that be better discussed here and used as a template for assays in this paper?

With our luciferase reporter strains in different *scuR/sarF* variants or treated with different sBI7 variants, we enriched our view on bacteriocin production. First, we showed that *slvX* and *slvY* are substantially overexpressed by the peptide sBI7 (Figure 4—figure supplement 2C). *blpK, slvV* and *slvW* promoters are also activated, albeit in a much lesser extent, in agreement with our data with the *scuR* overexpression mutant (Figure 2A). Finally, we showed that the sBI7-activated bacteriocin promoters are affected in a similar way when *scuR, sarF* or both genes were deleted (Figure 4—figure supplement 2C). We observed that the *sarF* deletion did not have a major impact on promoter activation, while the single *scuR* and double *scuR-sarF* deletion nearly or totally abolished it.

6) For the peptide docking and modeling data you need to provide the statistics of the model and docking fit provided by the programs you used.

The proposed model for the ScuR-sBI7 complex was not obtained by docking. Considering the high sequence conservation between ComR and ScuR, we used homology modelling to fit the sequence of ScuR on the structure of the peptide bound form of ComR from *S. thermophilus* and we replaced manually the sequence of the bound peptide observed in the crystal structure by the sequence of sBI7. The resulting ScuR-sBI7 model was optimized by a subsequent energy minimization run.